# BREAKING BEYOND COCO OBJECT DETECTION

## ABSTRACT

COCO dataset has become the de facto standard for training and evaluating object detectors. According to the recent benchmarks, however, performance on this dataset is still far from perfect, which raises the following questions, a) how far can we improve the accuracy on this dataset using deep learning, b) what is holding us back in making progress in object detection, and c) what are the limitations of the COCO dataset and how can they be mitigated. To answer these questions, first, we propose a systematic approach to determine the empirical upper bound in AP over *COCOval2017*, and show that this upper bound is significantly higher than the state-of-the-art mAP (78.2% *vs.* 58.8%). Second, we introduce two complementary datasets to COCO: i) COCO_OI, composed of images from COCO and OpenImages (from 80 classes in common) with 1,418,978 training bounding boxes over 380,111 images, and 41,893 validation bounding boxes over 18,299 images, and ii) ObjectNet_D containing objects in daily life situations (originally created for object recognition known as ObjectNet; 29 categories in common with COCO). We evaluate models on these datasets and pinpoint the annotation errors on the COCO validation set. Third, we characterize the sources of errors in modern object detectors using a recently proposed error analysis tool (TIDE) and find that models behave differently on these datasets compared to COCO. For instance, missing objects are more frequent in the new datasets. We also find that models lack out of distribution generalization. Code and data will be shared.

## 1 INTRODUCTION

Object recognition is believed, although debatable, to be solved in computer vision witnessed by the "superhuman" performance of the state-of-the-art (SOTA) models ($\sim$3% *vs.* $\sim$5% top-5 error rate human vs. machine on ImageNet (He et al., 2016; Dosovitskiy et al., 2020; Russakovsky et al., 2015)). Unlike object recognition, however, object detection remains largely unsolved and models perform far below the theoretcial upper bound (mAP=1). The best performance on *COCOval2017* and *COCOtest-dev2017* are 58.8% and 61%, respectively based on the COCO detection leaderboard & codalab results. According to the results from the 2019 OpenImages detection challenge, the best mAP on this dataset is 65.9%. Inspired by the recent stream of work such as Recht et al. (2019); Shankar et al. (2020); Beyer et al. (2020) examining the accuracy and out of distribution generalization of recognition models, we strive to understand why detection performance is poor and study the limitations of object detectors and the ways models and datasets can be improved.

Several years of extensive research in object detection has resulted in accumulation of an overwhelming amount of knowledge regarding model backbones, tips and tricks for model training, optimization, data collection, augmentation, annotation, model evaluation, and comparison, to a point that separating the wheat from the chaff is very difficult (Zou et al., 2019; Zhang et al., 2019). For example, getting all details right in implementing average precision (AP) is frustratingly difficult (see supplement). A quick Google search returns several blogs and codes with discrepant explanations of average precision (AP). In addition, it is not clear whether AP has started to saturate, whether a small improvement in AP (e.g., 56.1 vs. 56 mAP) is meaningful, and more importantly how much we can improve following the current trend, making one wonder maybe we have reached the peak of detection performance using deep learning[1]. A critical concern here is that maybe detection datasets are not big enough to capture variations in object size[2], viewpoint, occlusion, and spatial relationships

---

[1]This is also known as ceiling analysis.

[2]The median scale of the object relative to the image in ImageNet *vs.* COCO is 554 and 106, respectively. Therefore, most object instances in COCO are smaller than 1% of the image area (Singh & Davis, 2018).

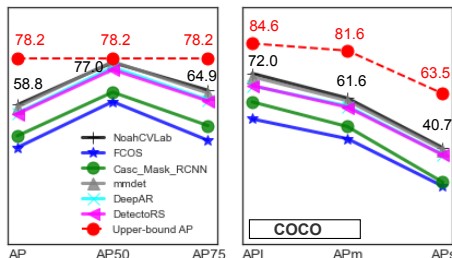

**Figure 1: Upper bound AP (UAP)** shown in red and scores of the best models using COCO evaluation tool (shown in black; numbers come from different models). Results over *COCOval2017* are compiled from the top entries of the latest challenge on COCO available at COCO detectoin leaderboard. See also paperswithcode.com coco_minival. Notice that the gap at $AP_{50}$ is almost closed in COCO. There is, however, still a large gap between best models and UAP. The gap is wider over higher IOUs and small objects. **Note:** Higher mAPs have been reported over the VOC benchmark paperswithcode.com VOC using the VOC evaluation tool (89.1 $AP_{50}$ Ghiasi et al. (2020) which is close to 89.5 UAP computed by us using the VOC tool; please see also Appx A).

among objects. In other words, scaling object detection seems to be much more challenging compared to scaling object recognition. Due to these, object detection can be considered as a key task to assess the promises and limits of deep learning in computer vision.

**Contributions.** To shed light on blind spots that could be holding back progress, we carefully and systematically approximate the empirical upper bound in AP (UAP). We argue that UAP is the score of the object detector with access to ground-truth bounding boxes. An object recognition model is trained on the training target bounding boxes and is then used to label the test target boxes (i.e., localization is assumed to be solved). In a nutshell, we find that there is a large gap between the best model mAP and the empirical upper bound as shown in Fig. 1. The gap is wider at higher IOUs and over small objects. Using the latest results on COCO dataset (Fig. 1), the gap is narrow at IOU=0.5 ($\sim$2 points). The computed UAP entails that there is a hope to reach this peak with the current tools if we can find smarter ways to adopt the object recognition models and backbones for object detection, but going beyond it may require major breakthroughs. The current trend in deep learning, in particular in recognition and detection, has been to scale the datasets. Less effort, however, has been spent to systematically inspect errors in datasets. Here, we perform such inspection, and investigate out of distribution generalization of object detectors by introducing new validation sets. We also introduce an extension of COCO by integrating it with OpenImages. Finally, we identify the bottlenecks in detectors and characterize the type of errors they make on these new sets.

## 2 RELATED RESEARCH

Four lines of work relate to our study. The first one includes **works that strive to understand object detectors, identify their shortcomings, and pinpoint where more research is needed**. Parikh & Zitnick (2011) aimed to find the weakest links in person detectors by replacing different components in a pipeline (e.g., part detection, non-maxima-suppression) with human annotations. Mottaghi et al. (2015) proposed human-machine CRFs for identifying bottlenecks in scene understanding models. Hoiem et al. (2012) inspected detection models in terms of their localization errors and confusion with other classes and the background over the PASCAL VOC dataset. They also conducted a meta-analysis to measure the impact of object properties such as color, texture, and real-world size on detection performance. To overcome the shortcomings of the Hoiem et al. 's approach and COCO analysis tools, recently Bolya et al. (2020) proposed to analyse the models by emphasizing the order in which the errors are considered. Russakovsky et al. (2013) analyzed the ImageNet localization task and emphasized on fine-grained recognition. Zhang et al. (2016) measured how far we are from solving pedestrian detection. Vondrick et al. (2013) proposed a method for visualizing object detection features to gain insights into their functioning. Some other related works in this regard include Li et al. (2019), Zhu et al. (2012), Zhang et al. (2014), Goldman et al. (2019), and Petsiuk et al. (2020).

The second line of work concerns research in **comparing object detection models**. Some works have analyzed and reported statistics and performances over benchmark datasets such as PASCAL VOC (Everingham et al., 2010; 2015), COCO (Lin et al., 2014), CityScapes (Cordts et al., 2016), and OpenImages (Kuznetsova et al., 2018). Recently, Huang et al. (2017) performed a speed/accuracy trade-off analysis of modern object detectors. Dollar et al. (2011) and Borji et al. (2015) compared

person detection and salient object detection models, respectively. Michaelis et al. (2019) assessed detection models on degraded images and observed about 30–60% performance drop, which could be mitigated by data augmentation. To resolve the issues with the AP score, some works have attempted to introduce alternative (e.g., Hall et al. (2018)) or complementary evaluation measures (e.g., Oksuz et al. (2018); Rezatofighi et al. (2019)).

Works in the third line study the **role of context in visual recognition** Bar (2004); Wolf & Bileschi (2006); Zhu et al. (2016); Marat & Itti (2012); Heitz & Koller (2008); Torralba & Sinha (2001); Rabinovich et al. (2007); Rosenfeld et al. (2018); Galleguillos & Belongie (2010). Heitz & Koller (2008) proposed a probabilistic framework to capture contextual information between "stuff" and "things" to improve detection. Barnea & Ben-Shahar (2019) utilized co-occurrence relations among objects to improve the detection scores. Divvala et al. (2009) explored the role of different types of context in recognition. Rosenfeld et al. (2018) showed that object detectors often miss objects that are transplanted out of their usual context (e.g., elephant in the living room). See also Heitz & Koller (2008), Chen et al. (2018), Song et al. (2011), Hu et al. (2018), Marat & Itti (2012), and Alamri & Pugeault (2019).

The fourth line of work deals with **generalization of object recognition models against synthetic and natural distribution shifts**. As an example, Recht et al. (2019) closely followed the original ImageNet creation process to build a new test set called ImageNetV2. They reported a performance gap of about 11% in top-1 accuracy between the performance of the best deep models on this dataset and the original test set. Similar observations have been made in Shankar et al. (2020); Beyer et al. (2020); Barbu et al. (2019). It is thus critical to ensure that models are learning meaningful generalizations, rather than just overfitting the idiosyncrasies in datasets. Towards this goal, a large number of works have assessed object recognition models and their robustness against image transformations and degradations such as spatial transformations, noise corruptions, simulated weather artifacts, and temporal changes. See for example Hendrycks & Dietterich (2019); Azulay & Weiss (2018); Recht et al. (2019); Mishkin et al. (2017); Gu et al. (2019).

## 3  ESTIMATING THE EMPIRICAL UPPER BOUND IN AP

We define the empirical upper bound in AP as the score of the object detector that a) **has access to the true location of the objects**, and b) **ground truth bounding boxes are labeled by the best object classifier**. This way we essentially assume that the localization problem is solved and what remains is only object recognition. Beware that we do not mean to undermine the importance of the localization component. What we intend to convey is that assuming no further progress in object recognition and investing all efforts in solving the localization problem can lead us to this upper bound and not beyond that. Knowing the upper bound can help us to better coordinate our efforts. The above object detector, however, may not give us the true upper bound AP due to the subtleties involved in AP calculation (Appx B). Specifically, it might be possible to improve upon this detector in at least two ways: a) by exploiting the local scene context around an object to improve the classification accuracy and thus better UAP, and/or b) by searching among the bounding boxes around the target object (those with a certain overlap with it) and see whether any of them can be classified better, compared to the target box itself. This does not matter for determining the UAP at the perfect IOU (=1) but may affect UAP at IOUs lower than one. We carefully investigate these details in the following.

### 3.1  EXPERIMENTAL SETUP

**Benchmarks and models.** We establish our analysis based on two recent large-scale object detection benchmarks: MMDetection (Chen et al., 2019b) and Detectron2 Wu et al. (2019). The former evaluates more than 50 models. The latter includes several variants of FastRCNN (Girshick, 2015). In both benchmarks, all COCO models have been trained on *train2017* and evaluated on *val2017*. We consider major object detection models including several variants of the RCNN such as FasterRCNN (Ren et al., 2015), MaskRCNN (He et al., 2017), RetinaNet (Lin et al., 2017), GridRCNN (Lu et al., 2019), LibraRCNN (Pang et al., 2019), CascadeRCNN (Cai & Vasconcelos, 2018), MaskScoringRCNN (Huang et al., 2019), GAFasterRCNN (Zhu et al., 2019), Hybrid Task Cascade (Chen et al., 2019a), and other models such as SSD (Liu et al., 2016), FCOS (Tian et al., 2019), and CenterNet (Zhou et al., 2019). Different backbones for each model are also taken into account. In addition, we also include the most recent models in the analysis.

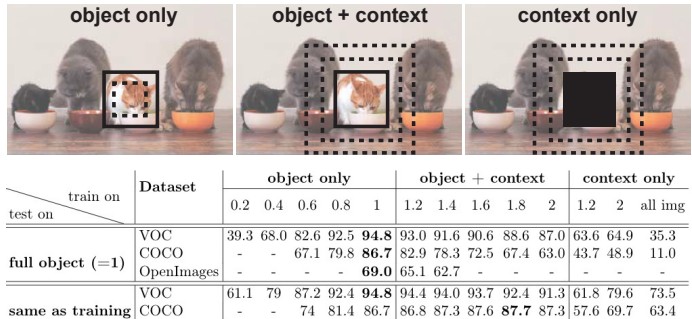

| object only | object + context | context only |
|---|---|---|

**Figure 2: Top:** Visual context around an object, **Bottom:** Object recognition accuracy. Top rows: testing on the canonical object size (=1 in the table, used in the rest of the paper; Appx D, ). Bottom rows: training and testing are the same (e.g., a classifier is trained on the object-only case 0.6 and is then tested on the object-only case 0.6). The best accuracy in each row is highlighted in **bold**.

| train on / test on | Dataset | object only | | | | | object + context | | | | | context only | | |
|---|---|---|---|---|---|---|---|---|---|---|---|---|---|---|
| | | 0.2 | 0.4 | 0.6 | 0.8 | 1 | 1.2 | 1.4 | 1.6 | 1.8 | 2 | 1.2 | 2 | all img |
| full object (=1) | VOC | 39.3 | 68.0 | 82.6 | 92.5 | **94.8** | 93.0 | 91.6 | 90.6 | 88.6 | 87.0 | 63.6 | 64.9 | 35.3 |
| | COCO | - | - | 67.1 | 79.8 | **86.7** | 82.9 | 78.3 | 72.5 | 67.4 | 63.0 | 43.7 | 48.9 | 11.0 |
| | OpenImages | - | - | - | - | **69.0** | 65.1 | 62.7 | - | - | - | - | - | - |
| same as training | VOC | 61.1 | 79 | 87.2 | 92.4 | **94.8** | 94.4 | 94.0 | 93.7 | 92.4 | 91.3 | 61.8 | 79.6 | 73.5 |
| | COCO | - | - | 74 | 81.4 | 86.7 | 86.8 | 87.3 | 87.6 | **87.7** | 87.3 | 57.6 | 69.7 | 63.4 |

**Datasets.** Three datasets including PASCAL VOC (Everingham et al., 2015), COCO (Lin et al., 2014), and OpenImages (Kuznetsova et al., 2018) are employed. Over VOC, we use *trainval0712* for training (16,551 images, 47,223 boxes) and *test2007* (4,952 images, 14,976 boxes) for testing. This dataset has 20 object categories. Over COCO, we use *train2017* for training (118,287 images, 860,001 boxes) and *val2017* (5,000 images, 36,781 boxes) for testing. COCO has carried the torch for benchmarking advances in object detection for the past 8 years. It has 80 object categories. Finally, we use OpenImages V4 dataset, used in the Kaggle competition. It has 500 object categories and contains 1,743,042 images (12,195,144 boxes) for training and 41,620 images (226,811 boxes) for validation. Its validation set is used here for testing.

## 3.2 UTILITY OF THE SURROUNDING CONTEXT

We trained ResNet152 (He et al., 2016) on target bounding boxes in 3 settings as shown in Fig. 2: 1) **object-only**, 2) **object+context**, and 3) **context-only**. Standard data augmentation techniques including color jittering, random horizontal flip, and random rotation (10 degrees) were applied. Boxes were resized to $224 \times 224$ pixels and models were trained for 15 epochs. Trained models were then tested on the target object boxes. Classification results (top-1 accuracy) are shown in the bottom panel of Fig. 2. We find that the canonical object size (full object) results in the best classification accuracy over all three datasets. Enlarging or shrinking the object bounding box lowers the performance. The context-only scenario results in a high classification score but still performs below other cases. Stretching the context to the whole scene drops the performance significantly.

## 3.3 SEARCHING FOR THE BEST LABEL & CONFIDENCE SCORE OF A TARGET BOUNDING BOX

Essentially the problem statement here is how best we can classify a target box by utilizing all available information in the scene. This is different from recognition models where objects are treated in isolation (see discussion section). Notice that recognition accuracy is not the same as AP since detection scores also matter in AP calculation (i.e., detections are ranked based on classification confidence). Having the best classifier in hand, we are ready to approximate the UAP.

**Strategies for labeling the boxes**. We explore two strategies in pursuit of UAP. In the **first one**, we apply the best classifier from the previous section to the canonical target bounding boxes. The detector built in this fashion gives the same AP regardless of the IOU threshold since our detections are target boxes. As we argued above, it is not possible to improve this detector at IOU=1. However, if we are interested in upper bound at a lower IOU threshold ($\gamma$), then it might be possible to do better by searching among the candidate boxes near a target box and choose the one that can be classified better than the target box, or by aggregating the information from all its nearby boxes. To account for this, in our **second strategy**, we sample boxes around an object, similar to Oksuz et al. (2020), and either use the original classifier (trained on the canonical object size) or train and test new classifiers on surrounding boxes. In any case, we always keep the target box but change its label and/or its classification confidence. First, let's take a look at our box sampling strategy illustrated in Fig. 3.

**Sampling boxes at IOU threshold $\geq \gamma$.** We are interested in finding the coordinates of the top-left corner of all rectangles[3] with IOU $\geq \gamma$ $(\gamma \leq 1)$ with the ground-truth bounding box. We use the coordinate system centered at the top-left corner $P$ of the target box (the PQRS rectangle; shown

---

[3]Here, we assume that all boxes have the same width and height as the target box. The solution can be easily extended to the case where rectangles are non-homogeneous.

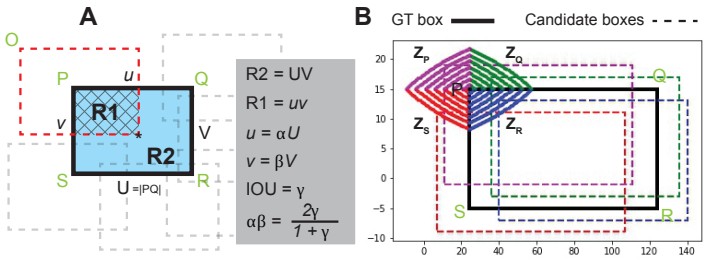

**Figure 3: A:** Illustration of our setup for finding boxes with IOU $\geq \gamma$ with the target box (corresponding to $\alpha\beta = 2\gamma/(1+\gamma)$; $\alpha\beta = 2/3$ for $IOU = 0.5$), **B:** The solutions are 4 curves represented by Eqs. 4 to 7. Four sample rectangles are shown with dashed lines.

in black) which can be easily converted to the image level coordinate frame. Let's first find the relationship between the coordinates of the point marked with * ($< u, v >$) and overlap threshold $\gamma$. According to the illustration in Fig. 3.A, we have:

$$R1 = uv, \quad R2 = UV, \quad \text{IOU} = \gamma, \quad \text{IOU} = \frac{R1}{2R2 - R1} \tag{1}$$

From these equations and assuming $u = \alpha U$ and $v = \beta V$, it is easy to derive the following equations:

$$R1 = \alpha U \beta V, \quad R1 = \frac{2\gamma}{1+\gamma} R2 \tag{2}$$

and from there we obtain:

$$\alpha\beta = \frac{2\gamma}{1+\gamma}, \quad (\alpha\beta = \frac{2}{3} \text{ for } \gamma = 0.5) \tag{3}$$

The same equation governs the coordinates of the bottom-left, top-left, and top-right corners of the rectangles intersecting with the target box at points $Q$, $R$, and $S$, respectively (in the coordinate frames centered at each of these points, in order). Calculating the top-left corner of these rectangles in their corresponding coordinate frames and representing them in the coordinate frame of the image, we arrive at the following four equations. Notice that these are curves and not lines.

$$Z_P: \quad \langle \ (\alpha - 1)U + x_P, \ (\beta - 1)V + y_P \ \rangle \tag{4}$$

$$Z_Q: \quad \langle \ (1 - \alpha)U + x_P, \ (\beta - 1)V + y_P \ \rangle \tag{5}$$

$$Z_R: \quad \langle \ (1 - \alpha)U + x_P, \ (1 - \beta)V + y_P \ \rangle \tag{6}$$

$$Z_S: \quad \langle \ (\alpha - 1)U + x_P, \ (1 - \beta)V + y_P \ \rangle \tag{7}$$

$$\forall \ \alpha, \beta \leq 1, \quad \text{s.t.} \quad \alpha\beta = \frac{2\gamma}{1+\gamma} \tag{8}$$

Using the above equations, we sample $m$ (here $m=4$) rectangles with $IOU \geq \gamma$ (Fig. 3.B) and label them with the label of the target box. We then train a new classifier (same ResNet152 as above) on these boxes. This is effectively a new data augmentation technique. Notice that UAP is the direct consequence of the classification

| Dataset | Acc. | Most Confident Box | | | | Most Frequent Label | | | |
|---------|------|------|--------|--------|--------|------|--------|--------|--------|
| | | AP | $AP_l$ | $AP_m$ | $AP_s$ | AP | $AP_l$ | $AP_m$ | $AP_s$ |
| VOC | 93.7 | 88.7 | 91.7 | 81.4 | 63.8 | 89.1 | 92.0 | 82.9 | 60 |
| COCO | 84.8 | 76.9 | 81.8 | 80.6 | 62.8 | 76.4 | 82.0 | 80.4 | 60.7 |

**Table 1:** Upper bound AP using our second strategy (i.e., searching for the best bounding box or object label near a target box; among boxes with IOU $\geq 0.5$) using a classifier trained and tested on surrounding boxes (see Appx C for results using the classifier trained on canonical objects). Notice that UAP, $UAP_{50}$, and $UAP_{75}$ are all the same.

accuracy, meaning if we can classify objects better, we can reach a higher UAP. To estimate UAP, we sample $m$ rectangles around a target box (with $IOU \geq \gamma$), and then label the target box with a) the label and confidence of the bounding box with the highest classification score (i.e., **the most confident box**), or b) the **most frequent label** among the nearby boxes and the maximum confidence score among them.

### 3.4 UPPER BOUND AP RESULTS

Here, we report classification scores, upper bound APs, the score of the models (mean AP over all IOUs; unless specified otherwise), and the breakdown AP over categories. See also Appxs E and F.

**Comparison of strategies.** Summary results of the first strategy are shown in Fig. 1. There is a large gap between the UAP and the theoretical upper bound, and between UAP and model mAP. Results of

the second strategy are presented in Table 3 using a classifier trained on surrounding boxes. Please see the Appx C for results using classifier trained on canonical objects and applied to surrounding boxes. Contrary to our expectation, the second strategy did not lead to higher UAPs perhaps because the surrounding boxes contain additional visual content which may introduce noise in the process. In other words, using surrounding boxes leads to less homogeneous samples (compared to using canonical objects) which hurts training. This leads to lower classification accuracy and hence a lower UAP. We found that sampling boxes at higher IOU thresholds (e.g., $\gamma = 0.9$) did not improve the results. Also, setting the confidence of the detections to 1 lowered the UAP. In what follows we only discuss the results using the first strategy.

**PASCAL VOC**. Fig. 22 shows results on VOC using the COCO evaluation tool. We trained and tested 5 models on this dataset including FasterRCNN, FCOS, SSD512, and two variants of CenterNet. The classification accuracy over VOC is very high about 94.8%. Consequently, UAP is high about 91.6%. FCOS does the best here with the mAP of 47.9% (dashed line). As can be seen, there is a large gap between the mAP of the best model and the UAP (~44 points). UAP is above model mAP in all categories. Models behave similarly across categories (i.e., low variance in each category). Results using the VOC evaluation code is given in Appx E. VOC code is based on IOU=0.5 and calculates the area under the PR curve in a slightly different way than the COCO code (See also Fig. 1).

**COCO**. Borrowing the *MMDetection* benchmark and adding the results from CenterNet to it, we end up comparing 15 models (71 in total; the combination of models and backbones). Among the models we tested, the best ones are Hybrid Task Cascade model (Chen et al., 2019a) and Cascade MaskRCNN (Cai & Vasconcelos, 2018) with 46.9% and 45.7% mAPs, respectively. See also Appx E and Appx F. The UAP on COCO is 78.2% which is about 35 points above the mAP of the best model tested here, and ~20 above the most recent model (see Fig. 1). Recall that UAP does not depend on the IOU threshold since detected boxes are ground-truth targets. The gap is much narrower at $AP_{50}$[4]. The gap between the UAP and model mAP over small objects is about 23 points which is almost 2x wider than the corresponding gap over large objects. The UAP over small objects is much lower than the UAP over large objects. This also holds for model mAPs.

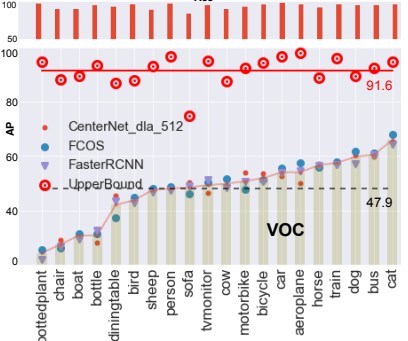

**Figure 4:** UAP and Model mAPs over the VOC dataset. Categories are sorted according to the average model AP. Top bar chart shows classification accuracy. Dashed line represents FCOS mAP. See also Appx E.

Breakdown APs over object categories are shown in Fig. 5. Here, we use the *Detectron2* benchmark which reports per-category APs mainly over RCNN model family. Among the 80 categories, only 3 (*snowboard, toothbrush, toaster*) have UAPs below the best model APs. We noticed that aggregate scores on *MMDetection* and *Detectron2* are quite consistent. Among 18 variants of FasterRCNN and MaskRCNN, the best model has the AP of 44.3 (dashed line in Fig. 1) which is lower than the best available model on COCO (58.8%; solid black line in Fig. 1) and the UAP (78.2%).

**OpenImages**. This dataset (Kuznetsova et al., 2018) is the latest endeavor in object detection and is much more challenging than its predecessors. Our classifier achieves 69.0% top-1 accuracy on the validation set of OpenImages V4 which is lower than the other three datasets.

**DOTA and Fashion datasets.** DOTA (here v1.0) is a dataset of aerial images and includes 15 categories Xia et al. (2018). The fashion dataset includes 40 classes of fashion items such as jeans, T-shirt, and backpack. Sample images from these datasets and additional results are shown in Appx. J. The UAPs over the datasets in order are 88% and 71.2%. The best model APs in order are 83.1% and 59.7%. We obtain the UAP of 58.9%, using the TensorFlow evaluation code for computing the AP score on this dataset, which is slightly different than the COCO AP calculation tool (here we discarded grouping and super-category). We are not aware of any model scores on this particular set of OpenImages V4.

---

[4]The best available scores on COCO*val2017* are shown in Fig. 1. Interestingly, the gap at $AP_{50}$ is almost closed (~1.2). There is, however, still a large gap at AP over all IOUs, and also over medium and small objects.

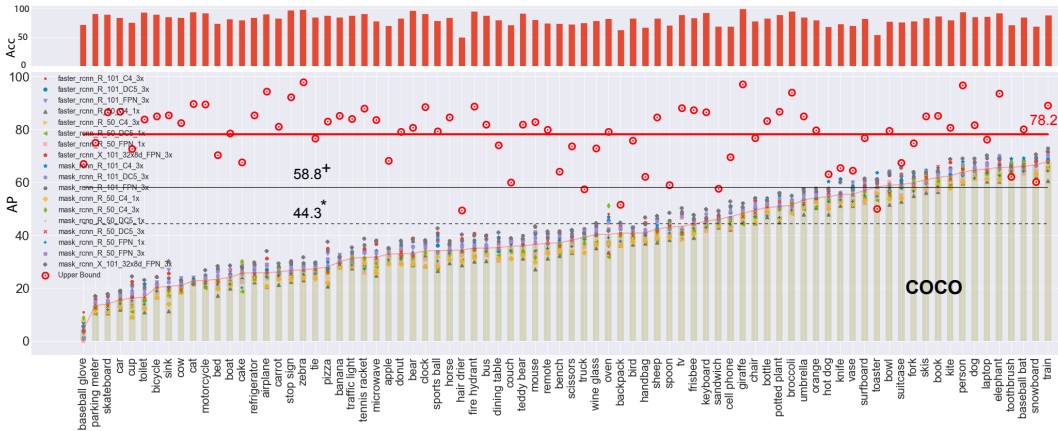

**Figure 5:** Detection APs over COCO dataset borrowed from the *Detectron2* benchmark. The black dash line corresponds to the best model among the models we analyzed (the score shown with "*"). The black solid line shows the most recent results (the score shown with "+"). See Appx E for results using *MMDetection*.

## 4 COMPLEMENTARY DATASETS TO COCO

The bulk of research on object detection has been primarily focused on developing network architectures or designing new loss functions; less attention has been devoted to data. A notable exception is OpenImages dataset. However, despite being orders of magnitude larger than COCO, it has been less adopted. To mitigate this, a) building on top of the COCO ecosystem, we extend the COCO dataset. We are motivated by studies (e.g., Dosovitskiy et al. (2020); Kolesnikov et al. (2019); Radford et al. (2021)) that have used very large datasets (e.g., JFT-300M Sun et al. (2017)) to achieve SOTA accuracy in object recognition (about 90% top-1 acc; 10% improvement over models trained on ImageNet). Our approach is orthogonal to works that have employed data augmentation to increase data (e.g., Zhang et al. (2019); Ghiasi et al. (2020)), and b) we inspect the quality of data in existing datasets. As datasets grow larger in size it becomes increasingly harder to maintain high quality. High quality data is crucial for proper assessment of model performance. It is also important to have multiple evaluation sets with different characteristics to gauge generalization and robustness of detectors. We construct new validation sets to evaluate detectors and diagnose model errors.

### 4.1 NEW DATASETS

**COCO_OI.** We selected images from all 80 categories of OpenImages that are in common with COCO, except `person`, `car`, `chair` classes which are 3 most frequent classes in COCO (Fig. 14; Appx H). This is to a) avoid making the dataset highly imbalanced[5], and b) keep the number of samples and computation for model training under control. Some classes from OpenImages fall under the same COCO category (e.g., `bear`, `brown bear`, `polar bear` all are mapped to the `bear` class in COCO). The class mappings are given in Appx G. To keep the image size in the range of COCO image size, images from OpenImages are resized such that the larger dimension is 640px while preserving the aspect ratio. The train set of COCO_OI has 1,418,978 annotations in total across 380,111 images which is larger than *COCOtrain2017* with 860K boxes and 118,287 images. From these, 261,824 images are selected from the train and test sets of OpenImages containing 558,977 boxes. As part of this dataset, we also create a validation set with 41,893 bounding boxes across 18,299 images chosen from validation set of OpenImages. See Fig. 6.

**ObjectNet_D.** We annotated the object bounding boxes in the ObjectNet dataset originally introduced by Barbu et al. (2019). This dataset contains objects in daily life situations (`https://objectnet.dev/`) and its images (50,000 across 313 categories) are pictured by Mechanical Turk workers using a mobile app in a variety of backgrounds, rotations, and imaging viewpoints. Barbu et al. reported a dramatic performance drop of the SOTA models on ObjectNet compared to their accuracy on ImageNet ($\sim$ 40-45% drop). Therefore, it would be interesting to see how well object detectors perform on this dataset. In total, 5875

---

[5]When discarding some object classes from OpenImages, any image with any object instance from these classes, even though it had some objects from other classes of interest, was discarded.

| Dataset | Model | AP | AP$_{50}$ | AP$_{75}$ | AP$_S$ | AP$_M$ | AP$_L$ |
|---------|-------|-----|------|------|------|------|------|
| COCO | Efficientdet | 51.2 | 70.4 | 55.1 | 35.7 | 55.8 | 64.8 |
|  | DetectoRS | 49.1 | 67.7 | 53.4 | 29.9 | 53 | 65.2 |
| COCO_OI | Efficientdet | 44 | 52.7 | 45.7 | 9 | 23.6 | 51.5 |
|  | DetectoRS | 51.5 | 51 | 45.1 | 11.4 | 25.5 | 50.9 |
| ObjectNet_D | Efficientdet | 23.9 | 41.6 | 24.5 | 0 | 5.7 | 24.7 |
|  | DetectoRS | 20.3 | 37.8 | 19.6 | 0 | 4.7 | 21.8 |

**Table 2:** Model performance over the validation sets of datasets.

bounding boxes across 5875 images (one annotated object per image) from 29 categories in common with COCO are annotated. Please see Appx H for more details on this dataset.

## 4.2 Model performance and error analysis

We tested two SOTA models, Efficientdet (Tan et al., 2019)[6] and DetectoRS (Cai & Vasconcelos, 2018) on the new validation sets (trained on COCO). As you can observe in Table 2, both models perform lower on ObjectNet_D and COCO_OI datasets compared to COCO. In particular, over ObjectNet_D, performance has been severely degraded partially because objects in this dataset are pictured in various, and sometimes odd, viewpoints and backgrounds (Fig. 6). Also, AP$_S$ and AP$_M$ on this dataset are both very low since almost all objects in ObjectNet_D are large, per the object size definitions in COCO.

To examine whether models behave similarly, we compare their errors over COCO and COCO_OI validation sets. To this end, we employ the recently proposed error diagnosis and analysis tool known as TIDE by Bolya et al. (2020). According to Fig. 7, both

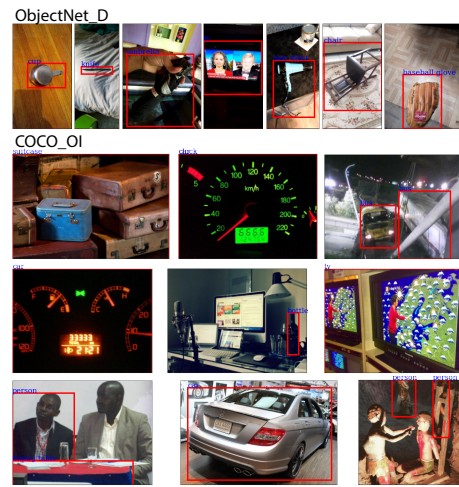

**Figure 6:** Samples from validation sets of ObjectNet_D and COCO_OI.

models behave the same on each dataset. Their behavior across datasets, however, is drastically different. On COCO, errors are almost equally distributed over 4 major error types (Loc, Cls, Bkg, and Miss), whereas over COCO_OI, Bkg (i.e., classifying background as an object) significantly outweighs other error types. This result signifies the impact of distribution shift on models.

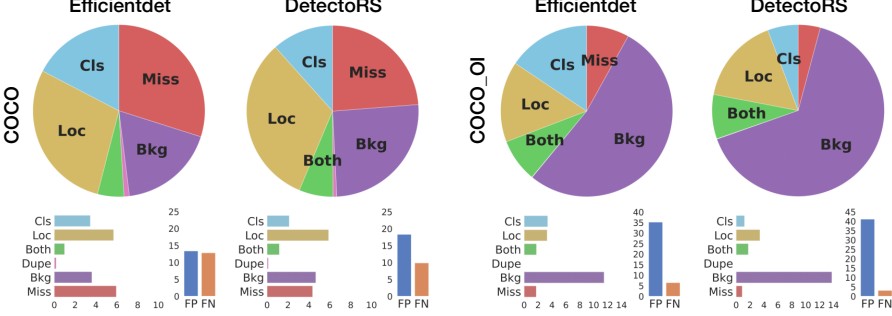

**Figure 7:** Error analysis over validation sets of COCO and COCO_OI datasets.

## 4.3 Limitations of the COCO dataset

To assess the quality of COCO annotations, we manually inspected *val2017* images, and observed that annotations have similar error types as in model predictions as illustrated in Fig. 8. We cluster the errors into four categories:

- **Misses**, where some objects are not annotated (e.g., the cup in Fig. 8.A). Missed objects are often peripheral ones, and are often small, occluded or cropped. Miss is the most frequent error type,
- **Non-existing objects or incorrect labels** are false positives due to either confusing background as an object (Bkg error type; e.g., the sheep in Fig. 8.B), or using a wrong label to annotate an object (Cls error type; i.e., confusion with other classes, e.g., goat labeled as cow in Fig. 8.B),

---

[6]efficientdet-d7.pth from https://github.com/zylo117/Yet-Another-EfficientDet-Pytorch

- **Localization error**, where bounding boxes around objects are not accurately drawn (Loc error type; Fig. 8.C),
- **Duplicates**, where duplicate overlapping bounding boxes are drawn around the same object (Dupe; Fig. 8.D). Also in some images a large bounding box covers multiple objects close to each other. These annotations are tagged with *iscrowd* in COCO (*group* in OpenImages) and are treated differently during evaluation. Moving forward it would be better to make these annotations more precise since they often contain a large portion of the image background. OpenImages dataset suffers from the same errors as in COCO. Some are shown in Fig. 6. ObjectNet_D does not have these mistakes by construction, but it is limited to only one annotated object per image (Fig. 6).

In addition to above, in Fig. 8.E we highlight some cases in which annotations are not consistent. For example, in some images, pictures of people in posters are annotated, wheres in some other images they are not. The same holds about reflections in the glass or in the mirror, as well as statues. In some images, an object is considered as part of another object and thus has been discarded (e.g., keyboard on laptop; last two images in Fig. 8.E). Finally, some objects are very hard to detect even for humans making the annotations debatable (i.e., low annotation consistency among people; Fig. 8.F). For instance, it is not clear whether the right-most boxes in the first image contain cars, or the animal on the right-side of the third image is actually a zebra. Images and annotations of this sort can be either discarded or treated differently when evaluating models. In total, we found that over COCOval2017, 52 objects are missed, 67 objects have incorrect labels, 89 objects have imprecise bounding boxes, and 19 objects have duplicate bounding boxes. We will share the corrected annotations.

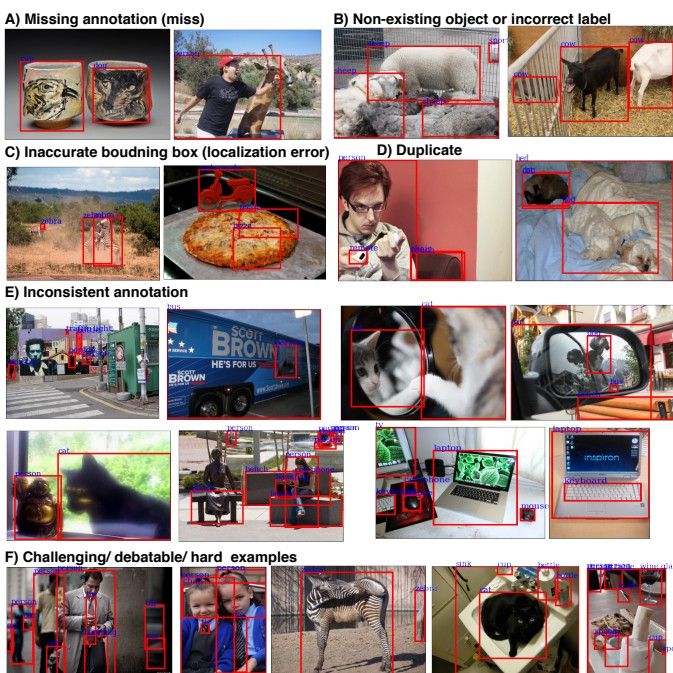

**Figure 8:** Some annotation errors on *COCOval2017*. See also Appx I.

## 5 DISCUSSIONS, LIMITATIONS, AND FUTURE WORK

We found that the best detectors still underperform what is empirically possible, and the performance gap is wider over small objects. In measuring UAP, we label target boxes using a trained classifier on objects. This essentially means that the deficit in mAP is mainly due to the classification error since other error types are fixed. Our approach is complementary to error analysis tools and provides additional insights into the bottlenecks in object detection from a model-independent perspective.

We foresee the following directions for future research: a) using better classifiers (e.g., transformers Dosovitskiy et al. (2020)) or ensembles of models to estimate the UAP, b) using the proposed techniques to generate even larger training datasets which can open up more research opportunities for robustness, transfer learning, and semi-supervised learning in object detection, c) employing COCO_OI for training models and comparing the results with models trained on COCO, d) fixing errors in detection datasets and re-evaluating models on them, e) analysing whether and how models utilize visual context in object detection, whether they do it similar to humans (Rosenfeld et al., 2018; Barnea & Ben-Shahar, 2019; Singh et al., 2020), and whether results are consistent with studies of context prior to deep learning era (e.g., Rabinovich et al. (2007)), and f) invariance and robustness analysis of object detectors with respect to natural (e.g., domain shift) and synthetic distribution shifts (e.g., geometric transformations, blur).

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

## A  STATE OF THE ART IN OBJECT DETECTION PERFORMANCE

Fig. 9 shows the state of the art in object detection performance over VOC and COCO datasets taken from `https://paperswithcode.com/`. It also shows our computed upper bound mAP (thick black lines). While there has been a steady progress over time, there is still a significant gap between models and what is empirically possible.

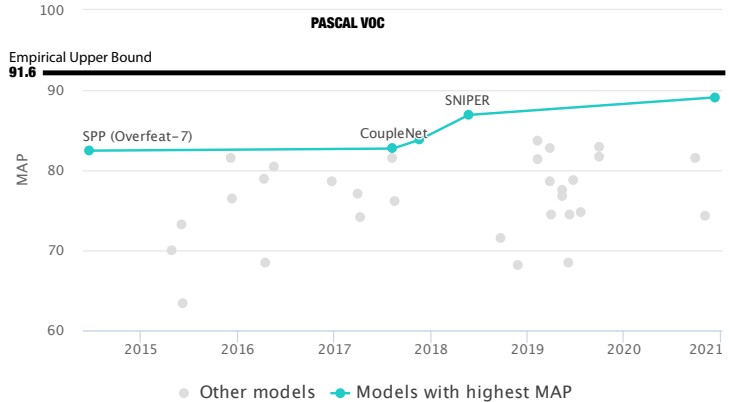

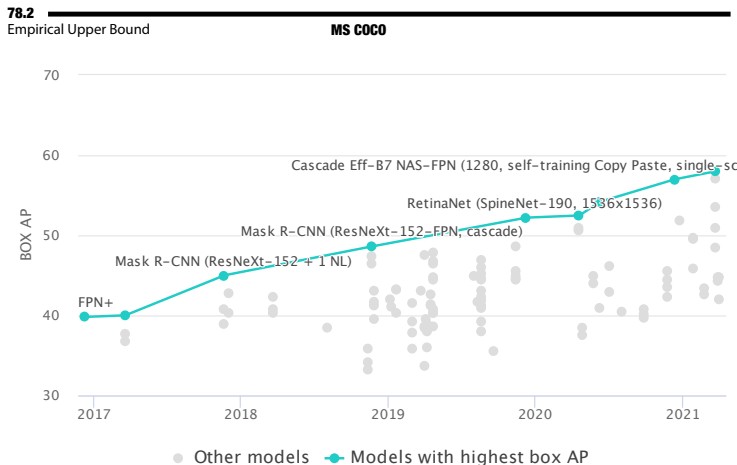

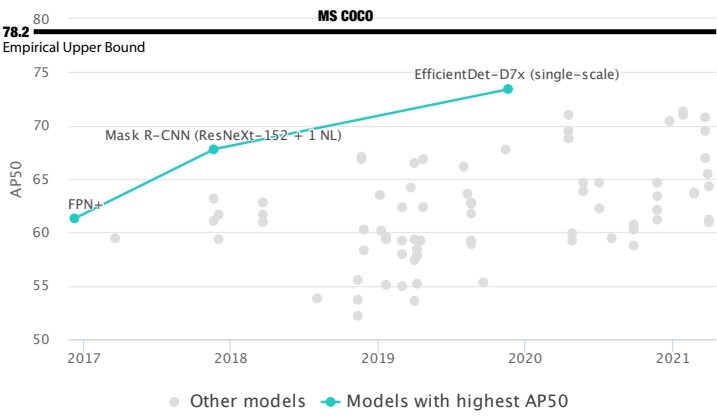

**Figure 9:** State of the art object detection performance taken from `https://paperswithcode.com/` as well as our computed upper bound mAP.

## B   AP CALCULATION

AP calculation compiled from here:

1. For the entire dataset, sort all the detections labeled with the same class with respect to their confidence scores.

2. Now, go over these sorted detection list and check whether each detection can be assigned to a ground truth. The assignment (or labeling as a True Positive) is based on Intersection Over Union (IoU) of the detection with a ground truth. There is a true positive validation threshold in terms of IoU and it is generally 0.5. In this step, note that a ground truth can be matched with only one detection (and this detection is the one with the highest confidence score since we go over a sorted list).

3. At the end of the previous step, we have identified the True Positive (TP) detection boxes, False Positive (FP) detection boxes and False Negative (FN) ground truth boxes. So, by going over the sorted detection list again, we can find precision for each recall to draw the Recall-Precision (RP) curve. The blue curve in the figure is the RP curve.

4. At the end of the previous step, we have identified the True Positive (TP) detection boxes, False Positive (FP) detection boxes and False Negative (FN) ground truth boxes. So, by going over the sorted detection list again, we can find precision for each recall to draw the Recall-Precision (RP) curve. The blue curve in the figure is the RP curve.

5. To discard the wiggles of the RP curve, at some recall point interpolate the blue curve to the highest precision possible at the positive side of this recall point. Thus, the red curve, called the interpolated RP curve, is obtained.

## C   RESULTS OF OUR SECOND STRATEGY FOR COMPUTING UAP

Here, we present the AP results of our second strategy using the original classifier trained on the canonical object size and applied to the surrounding boxes (instead of training a new classifier on the surrounding boxes). In the main text we reported results using our first strategy which is using the same canonical object size during training and testing (i.e., no search for the best surrounding box).

| Dataset | Acc. | Most Confident Box | | | | Most Frequent Label | | | |
|---|---|---|---|---|---|---|---|---|---|
| | | $AP$ | $AP_l$ | $AP_m$ | $AP_s$ | $AP$ | $AP_l$ | $AP_m$ | $AP_s$ |
| VOC | 91.6 | 88.7 | 91.7 | 81.4 | 63.8 | 89.1 | 92.0 | 82.9 | 60 |
| COCO | 82.6 | 76.9 | 81.8 | 80.6 | 62.8 | 76.4 | 82.0 | 80.4 | 60.7 |

**Table 3:** Results of our second strategy for estimating the upper-bound AP (i.e., searching for the best bounding box or object label near a target box; among boxes with IOU $\geq 0.5$). Here, we have used the original classifier trained on the canonical object size to classify the surrounding boxes, instead of training new ones on the surrounding boxes.

## D   CONFUSION MATRICES

These are the confusion matrices associated with the classifier trained on the canonical objects results of which is presented in Fig. 2 in the main text (i.e., our first strategy).

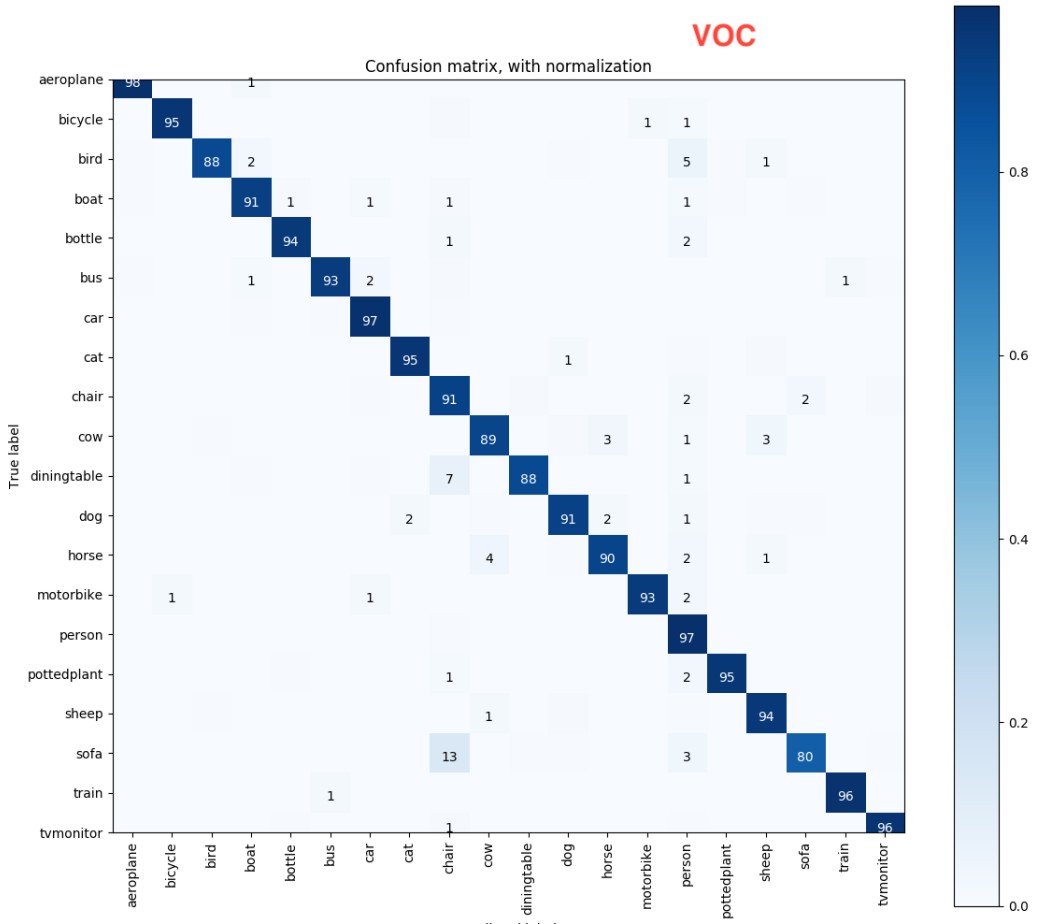

**Figure 10:** Confusion matrix of the trained classifier on the original object size over the VOC dataset (numbers are in %).

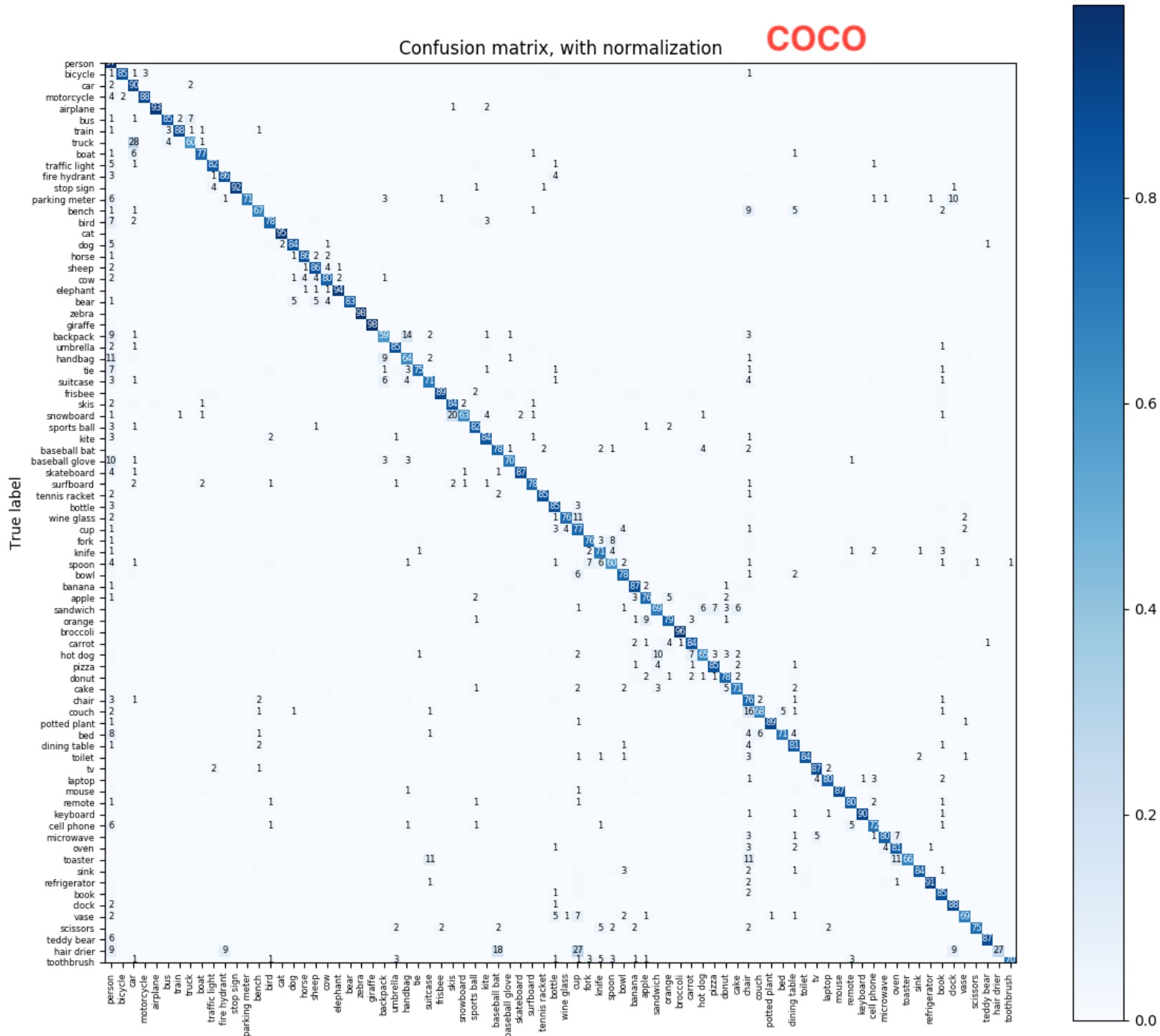

**Figure 11:** Confusion matrix of the trained classifier on the original object size over the COCO dataset (numbers are in %).

# E    MODEL AP AND UAP OVER PASCAL VOC AND COCO DATASETS

UAP over VOC using the COCO and VOC evaluation codes is shown in Fig. 12. In 13, we show results using the MMDetection benchmark for object detection over COCO using the COCO evaluation code.

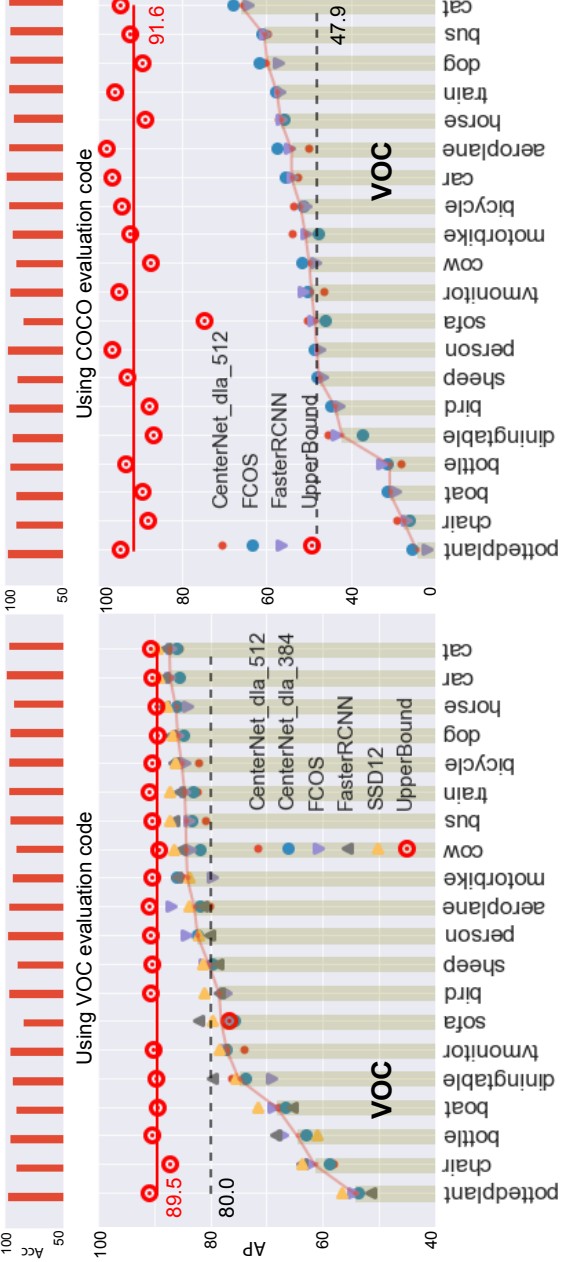

**Figure 12:** Model scores and upper bound AP over PASCAL VOC dataset using VOC (left) and COCO AP evaluation codes (right). Categories are sorted based on the average model AP. Bar charts show classification scores. Solid red and dashed black lines represent upper bound AP, and the best model AP, respectively.

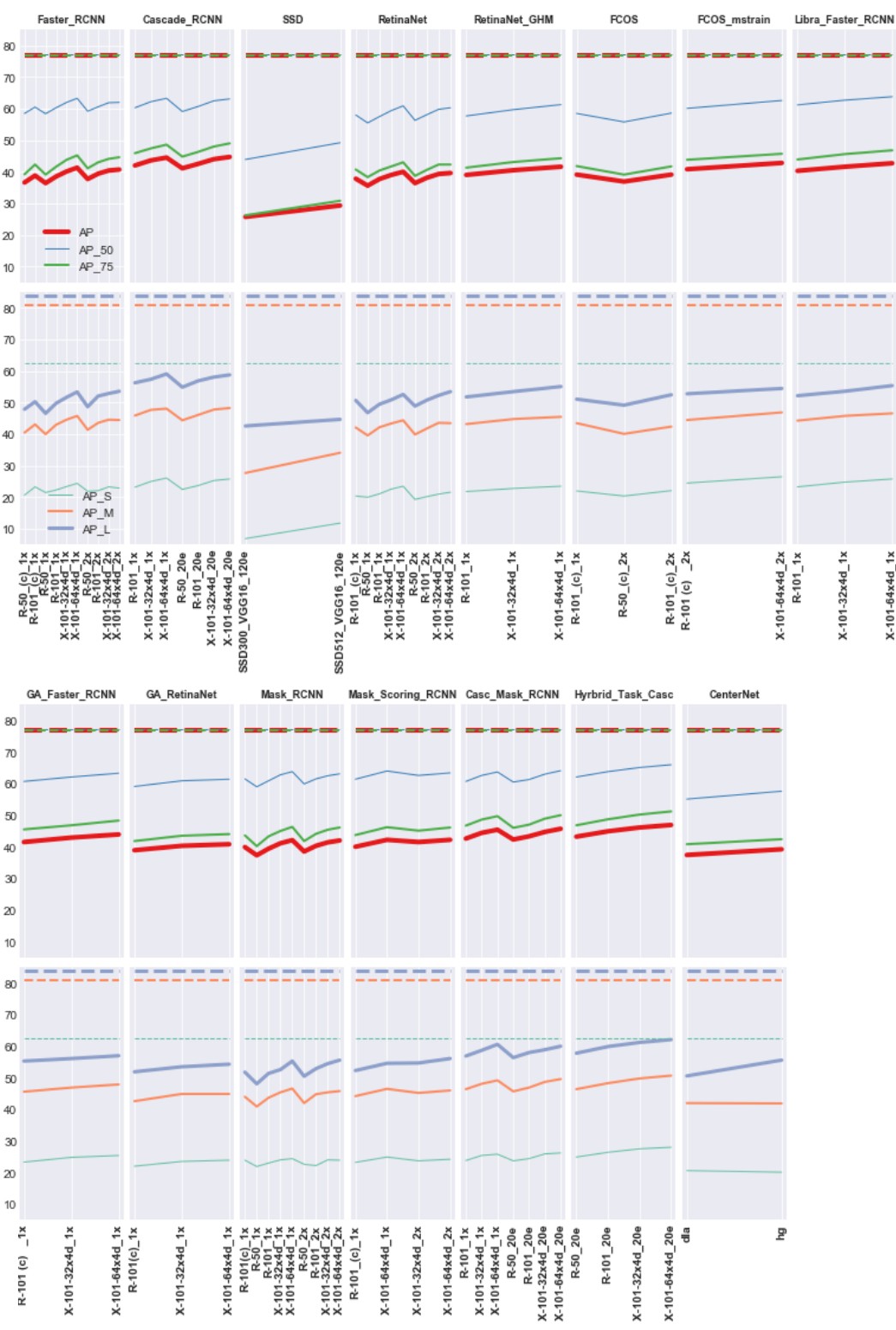

**Figure 13:** APs over COCO dataset borrowed from the *MMDetection* benchmark. We add CenterNet results to *MMDetection*.

# F    RAW MODEL AP AND UAP PERFORMANCES OVER VOC AND COCO DATASETS

Here, we show the detailed performance of UAP and AP of object detectors over COCO and VOC datasets results of which presented throughout the paper. In addition to precision, recall is also presented.

```
% ----------------  Upper-bound AP results  -------------

% COCO dataset
~~~~ Summary metrics ~~~~
Average Precision  (AP) @[ IoU=0.50:0.95 | area=   all | maxDets=100 ] = 0.782
Average Precision  (AP) @[ IoU=0.50      | area=   all | maxDets=100 ] = 0.782
Average Precision  (AP) @[ IoU=0.75      | area=   all | maxDets=100 ] = 0.782
Average Precision  (AP) @[ IoU=0.50:0.95 | area= small | maxDets=100 ] = 0.635
Average Precision  (AP) @[ IoU=0.50:0.95 | area=medium | maxDets=100 ] = 0.816
Average Precision  (AP) @[ IoU=0.50:0.95 | area= large | maxDets=100 ] = 0.846
Average Recall     (AR) @[ IoU=0.50:0.95 | area=   all | maxDets=  1 ] = 0.483
Average Recall     (AR) @[ IoU=0.50:0.95 | area=   all | maxDets= 10 ] = 0.797
Average Recall     (AR) @[ IoU=0.50:0.95 | area=   all | maxDets=100 ] = 0.812
Average Recall     (AR) @[ IoU=0.50:0.95 | area= small | maxDets=100 ] = 0.663
Average Recall     (AR) @[ IoU=0.50:0.95 | area=medium | maxDets=100 ] = 0.843
Average Recall     (AR) @[ IoU=0.50:0.95 | area= large | maxDets=100 ] = 0.893

% PASCAL VOC dataset
~~~~ Summary metrics ~~~~
Average Precision  (AP) @[ IoU=0.50:0.95 | area=   all | maxDets=100 ] = 0.916
Average Precision  (AP) @[ IoU=0.50      | area=   all | maxDets=100 ] = 0.916
Average Precision  (AP) @[ IoU=0.75      | area=   all | maxDets=100 ] = 0.916
Average Precision  (AP) @[ IoU=0.50:0.95 | area= small | maxDets=100 ] = 0.707
Average Precision  (AP) @[ IoU=0.50:0.95 | area=medium | maxDets=100 ] = 0.861
Average Precision  (AP) @[ IoU=0.50:0.95 | area= large | maxDets=100 ] = 0.941
Average Recall     (AR) @[ IoU=0.50:0.95 | area=   all | maxDets=  1 ] = 0.579
Average Recall     (AR) @[ IoU=0.50:0.95 | area=   all | maxDets= 10 ] = 0.908
Average Recall     (AR) @[ IoU=0.50:0.95 | area=   all | maxDets=100 ] = 0.930
Average Recall     (AR) @[ IoU=0.50:0.95 | area= small | maxDets=100 ] = 0.736
Average Recall     (AR) @[ IoU=0.50:0.95 | area=medium | maxDets=100 ] = 0.877
Average Recall     (AR) @[ IoU=0.50:0.95 | area= large | maxDets=100 ] = 0.954

% ----------------  Model performance  -------------

% ----------------  COCO dataset  -------------

% FasterRCNN
~~~~ Summary metrics ~~~~
Average Precision  (AP) @[ IoU=0.50:0.95 | area=   all | maxDets=100 ] = 0.364
Average Precision  (AP) @[ IoU=0.50      | area=   all | maxDets=100 ] = 0.584
Average Precision  (AP) @[ IoU=0.75      | area=   all | maxDets=100 ] = 0.391
Average Precision  (AP) @[ IoU=0.50:0.95 | area= small | maxDets=100 ] = 0.215
Average Precision  (AP) @[ IoU=0.50:0.95 | area=medium | maxDets=100 ] = 0.400
Average Precision  (AP) @[ IoU=0.50:0.95 | area= large | maxDets=100 ] = 0.466
Average Recall     (AR) @[ IoU=0.50:0.95 | area=   all | maxDets=  1 ] = 0.304
Average Recall     (AR) @[ IoU=0.50:0.95 | area=   all | maxDets= 10 ] = 0.489
Average Recall     (AR) @[ IoU=0.50:0.95 | area=   all | maxDets=100 ] = 0.514
Average Recall     (AR) @[ IoU=0.50:0.95 | area= small | maxDets=100 ] = 0.324
Average Recall     (AR) @[ IoU=0.50:0.95 | area=medium | maxDets=100 ] = 0.554
```

```
Average Recall       (AR) @[ IoU=0.50:0.95 | area= large | maxDets=100 ] = 0.645

% FCOS
~~~~ Summary metrics ~~~~
Average Precision  (AP) @[ IoU=0.50:0.95 | area=   all | maxDets=100 ] = 0.428
Average Precision  (AP) @[ IoU=0.50      | area=   all | maxDets=100 ] = 0.626
Average Precision  (AP) @[ IoU=0.75      | area=   all | maxDets=100 ] = 0.457
Average Precision  (AP) @[ IoU=0.50:0.95 | area= small | maxDets=100 ] = 0.265
Average Precision  (AP) @[ IoU=0.50:0.95 | area=medium | maxDets=100 ] = 0.469
Average Precision  (AP) @[ IoU=0.50:0.95 | area= large | maxDets=100 ] = 0.545
Average Recall     (AR) @[ IoU=0.50:0.95 | area=   all | maxDets=  1 ] = 0.345
Average Recall     (AR) @[ IoU=0.50:0.95 | area=   all | maxDets= 10 ] = 0.552
Average Recall     (AR) @[ IoU=0.50:0.95 | area=   all | maxDets=100 ] = 0.582
Average Recall     (AR) @[ IoU=0.50:0.95 | area= small | maxDets=100 ] = 0.388
Average Recall     (AR) @[ IoU=0.50:0.95 | area=medium | maxDets=100 ] = 0.628
Average Recall     (AR) @[ IoU=0.50:0.95 | area= large | maxDets=100 ] = 0.735

% RetinaNet
~~~~ Summary metrics ~~~~
Average Precision  (AP) @[ IoU=0.50:0.95 | area=   all | maxDets=100 ] = 0.400
Average Precision  (AP) @[ IoU=0.50      | area=   all | maxDets=100 ] = 0.609
Average Precision  (AP) @[ IoU=0.75      | area=   all | maxDets=100 ] = 0.430
Average Precision  (AP) @[ IoU=0.50:0.95 | area= small | maxDets=100 ] = 0.235
Average Precision  (AP) @[ IoU=0.50:0.95 | area=medium | maxDets=100 ] = 0.444
Average Precision  (AP) @[ IoU=0.50:0.95 | area= large | maxDets=100 ] = 0.526
Average Recall     (AR) @[ IoU=0.50:0.95 | area=   all | maxDets=  1 ] = 0.328
Average Recall     (AR) @[ IoU=0.50:0.95 | area=   all | maxDets= 10 ] = 0.522
Average Recall     (AR) @[ IoU=0.50:0.95 | area=   all | maxDets=100 ] = 0.555
Average Recall     (AR) @[ IoU=0.50:0.95 | area= small | maxDets=100 ] = 0.361
Average Recall     (AR) @[ IoU=0.50:0.95 | area=medium | maxDets=100 ] = 0.599
Average Recall     (AR) @[ IoU=0.50:0.95 | area= large | maxDets=100 ] = 0.704

% SSD
~~~~ Summary metrics ~~~~
Average Precision  (AP) @[ IoU=0.50:0.95 | area=   all | maxDets=100 ] = 0.293
Average Precision  (AP) @[ IoU=0.50      | area=   all | maxDets=100 ] = 0.492
Average Precision  (AP) @[ IoU=0.75      | area=   all | maxDets=100 ] = 0.308
Average Precision  (AP) @[ IoU=0.50:0.95 | area= small | maxDets=100 ] = 0.118
Average Precision  (AP) @[ IoU=0.50:0.95 | area=medium | maxDets=100 ] = 0.341
Average Precision  (AP) @[ IoU=0.50:0.95 | area= large | maxDets=100 ] = 0.447
Average Recall     (AR) @[ IoU=0.50:0.95 | area=   all | maxDets=  1 ] = 0.264
Average Recall     (AR) @[ IoU=0.50:0.95 | area=   all | maxDets= 10 ] = 0.400
Average Recall     (AR) @[ IoU=0.50:0.95 | area=   all | maxDets=100 ] = 0.425
Average Recall     (AR) @[ IoU=0.50:0.95 | area= small | maxDets=100 ] = 0.173
Average Recall     (AR) @[ IoU=0.50:0.95 | area=medium | maxDets=100 ] = 0.488
Average Recall     (AR) @[ IoU=0.50:0.95 | area= large | maxDets=100 ] = 0.607
```

```
% ----------------  VOC results using COCO evaluation code -------------

% CenterNet
Average Precision  (AP) @[ IoU=50:95 | area=   all | maxDets=100 ] = 47.8
Average Precision  (AP) @[ IoU=50      | area=   all | maxDets=100 ] = 72.7
Average Precision  (AP) @[ IoU=75      | area=   all | maxDets=100 ] = 51.3
Average Precision  (AP) @[ IoU=50:95 | area= small | maxDets=100 ] = 07.4
Average Precision  (AP) @[ IoU=50:95 | area=medium | maxDets=100 ] = 26.2
Average Precision  (AP) @[ IoU=50:95 | area= large | maxDets=100 ] = 61.0
Average Recall     (AR) @[ IoU=50:95 | area=   all | maxDets=  1 ] = 40.2
Average Recall     (AR) @[ IoU=50:95 | area=   all | maxDets= 10 ] = 57.2
Average Recall     (AR) @[ IoU=50:95 | area=   all | maxDets=100 ] = 58.4
Average Recall     (AR) @[ IoU=50:95 | area= small | maxDets=100 ] = 18.6
Average Recall     (AR) @[ IoU=50:95 | area=medium | maxDets=100 ] = 40.9
Average Recall     (AR) @[ IoU=50:95 | area= large | maxDets=100 ] = 70.5

% FCOS
Average Precision  (AP) @[ IoU=50:95 | area=   all | maxDets=100 ] = 47.9
Average Precision  (AP) @[ IoU=50      | area=   all | maxDets=100 ] = 71.0
Average Precision  (AP) @[ IoU=75      | area=   all | maxDets=100 ] = 51.4
Average Precision  (AP) @[ IoU=50:95 | area= small | maxDets=100 ] = 11.1
Average Precision  (AP) @[ IoU=50:95 | area=medium | maxDets=100 ] = 32.1
Average Precision  (AP) @[ IoU=50:95 | area= large | maxDets=100 ] = 58.4
Average Recall     (AR) @[ IoU=50:95 | area=   all | maxDets=  1 ] = 41.2
Average Recall     (AR) @[ IoU=50:95 | area=   all | maxDets= 10 ] = 58.5
Average Recall     (AR) @[ IoU=50:95 | area=   all | maxDets=100 ] = 59.5
Average Recall     (AR) @[ IoU=50:95 | area= small | maxDets=100 ] = 19.5
Average Recall     (AR) @[ IoU=50:95 | area=medium | maxDets=100 ] = 45.2
Average Recall     (AR) @[ IoU=50:95 | area= large | maxDets=100 ] = 70.2

% MASK RCNN  / FasterRCNN
Average Precision  (AP) @[ IoU=50:95 | area=   all | maxDets=100 ] = 47.3
Average Precision  (AP) @[ IoU=50      | area=   all | maxDets=100 ] = 71.3
Average Precision  (AP) @[ IoU=75      | area=   all | maxDets=100 ] = 52.6
Average Precision  (AP) @[ IoU=50:95 | area= small | maxDets=100 ] = 08.6
Average Precision  (AP) @[ IoU=50:95 | area=medium | maxDets=100 ] = 30.7
Average Precision  (AP) @[ IoU=50:95 | area= large | maxDets=100 ] = 58.1
Average Recall     (AR) @[ IoU=50:95 | area=   all | maxDets=  1 ] = 40.3
Average Recall     (AR) @[ IoU=50:95 | area=   all | maxDets= 10 ] = 53.8
Average Recall     (AR) @[ IoU=50:95 | area=   all | maxDets=100 ] = 54.1
Average Recall     (AR) @[ IoU=50:95 | area= small | maxDets=100 ] = 11.2
Average Recall     (AR) @[ IoU=50:95 | area=medium | maxDets=100 ] = 36.9
Average Recall     (AR) @[ IoU=50:95 | area= large | maxDets=100 ] = 65.7

% -------  VOC results over categories using VOC evaluation code  -------

% Upper bound

AP for aeroplane = 0.9091
AP for bicycle = 0.9033
AP for bird = 0.9065
AP for boat = 0.8951
AP for bottle = 0.9056
AP for bus = 0.9039
AP for car = 0.9052
```

```
AP for cat = 0.9062
AP for chair = 0.8722
AP for cow = 0.8933
AP for diningtable = 0.8968
AP for dog = 0.8950
AP for horse = 0.8969
AP for motorbike = 0.9054
AP for person = 0.9066
AP for pottedplant = 0.9085
AP for sheep = 0.9035
AP for sofa = 0.7672
AP for train = 0.9087
AP for tvmonitor = 0.9012
Mean AP = 0.8945

% FCOS

AP for aeroplane = 0.8701
AP for bicycle = 0.8454
AP for bird = 0.7722
AP for boat = 0.6895
AP for bottle = 0.6709
AP for bus = 0.8371
AP for car = 0.8716
AP for cat = 0.8704
AP for chair = 0.6213
AP for cow = 0.8362
AP for diningtable = 0.6900
AP for dog = 0.8572
AP for horse = 0.8414
AP for motorbike = 0.7966
AP for person = 0.8430
AP for pottedplant = 0.5464
AP for sheep = 0.8107
AP for sofa = 0.7679
AP for train = 0.8454
AP for tvmonitor = 0.7735
Mean AP = 0.7829

% FasterRcnn

AP for aeroplane = 0.8147
AP for bicycle = 0.8656
AP for bird = 0.7855
AP for boat = 0.6552
AP for bottle = 0.6848
AP for bus = 0.8660
AP for car = 0.8821
AP for cat = 0.8811
AP for chair = 0.6376
AP for cow = 0.8527
AP for diningtable = 0.7977
AP for dog = 0.8691
AP for horse = 0.8816
AP for motorbike = 0.8602
```

```
AP for person = 0.8008
AP for pottedplant = 0.5136
AP for sheep = 0.7870
AP for sofa = 0.8221
AP for train = 0.8543
AP for tvmonitor = 0.7778
Mean AP = 0.7945

% SSD12

AP for aeroplane = 0.8388
AP for bicycle = 0.8619
AP for bird = 0.8119
AP for boat = 0.7161
AP for bottle = 0.6090
AP for bus = 0.8730
AP for car = 0.8910
AP for cat = 0.8951
AP for chair = 0.6361
AP for cow = 0.8652
AP for diningtable = 0.7550
AP for dog = 0.8681
AP for horse = 0.8772
AP for motorbike = 0.8373
AP for person = 0.8215
AP for pottedplant = 0.5666
AP for sheep = 0.8127
AP for sofa = 0.7970
AP for train = 0.8723
AP for tvmonitor = 0.7855
Mean AP = 0.7996
```

## G  LIST OF OBJECT CATEGORIES IN NEW DATASETS

The list of object categories present in the COCO along with their corresponding classes in the OpenImages dataset is given below. The mapping between COCO to ObjectNet_$D$ datasets is also presented. Note that some classes in COCO correspond to more than one class in the destination dataset (i.e., one to many relationship).

**COCO to OpenImages category mapping:**

```
COCO_to_OI_dict = {'Baseball glove' : 'Baseball glove',
'Parking meter' : 'Parking meter',
'Skateboard' : 'Skateboard',
'Car' : 'Car',
'Cup' : ['coffee cup', 'Measuring cup'],
'Toilet' : 'Toilet',
'bicycle' : 'bicycle',
'sink' : 'sink',
'cow': ['Cattle', 'Bull'],
'cat': 'cat',
'Motorcycle' : 'Motorcycle',
'Bed' : 'Bed',
'boat' : 'boat',
'Cake' : 'Cake',
'refrigerator' : 'refrigerator',
'Airplane' : 'Airplane',
```

```
'Carrot' : 'Carrot',
'Stop sign' : 'Stop sign',
'Zebra' : 'Zebra',
'Tie' : 'Tie',
'Pizza' : 'Pizza',
'Banana' : 'Banana',
'Traffic light' : 'Traffic light',
'Tennis racket' : 'Tennis racket',
'Microwave' : 'Microwave oven',
'Apple': 'Apple',
'donut': 'Doughnut',
'Bear': ['Bear', 'Brown bear', 'Polar bear'],
'Teddy bear' : 'Teddy bear',
'clock': ['Alarm clock', 'Clock', 'Digital clock'],
'sports ball' : ['Ball', 'Football', 'Volleyball' , 'Tennis Ball',
                  'Rugby ball', 'golf ball',  'Cricket ball'],
'horse': 'horse',
'Hair drier': 'Hair dryer',
'Fire hydrant': 'Fire hydrant',
'Bus': 'Bus',
'Dining Table' : ['Table', 'Kitchen & dining room table'],
'Couch': ['Couch', 'Studio couch'],
'mouse': 'computer mouse',
'Remote': 'Remote control',
'Bench': 'Bench',
'Scissors': 'Scissors',
'Truck': 'Truck',
'Wine glass': 'Wine glass',
'Oven': 'Oven',
'Backpack': 'Backpack',
'Bird': 'Bird',
'Handbag': 'Handbag',
'Sheep': 'Sheep',
'Spoon': 'Spoon',
'Tv':'Television',
'Frisbee' : 'Flying disc',
'Keyboard' : 'computer Keyboard',
'Sandwich' : ['Sandwich', 'Submarine sandwich'],
'Cell phone' : 'Mobile phone',
'Giraffe' : 'Giraffe',
'Chair' : 'Chair',
'Bottle' : 'Bottle',
'Potted plant' : 'Houseplant',
'Broccoli' : 'Broccoli',
'Umbrella' : 'Umbrella',
'Orange' : 'Orange',
'Hot dog' : 'Hot dog',
'Knife' : 'kitchen knife',
'Vase' : 'Vase',
'Surfboard' : 'Surfboard',
'Toaster' : 'Toaster',
'Bowl' : ['bowl', 'Mixing bowl'],
'Suitcase' : 'Suitcase',
'Fork' : 'Fork',
'Skis' : 'Ski',
'Book' : 'Book',
'Kite' : 'Kite',
'Person' : 'Person',
'Dog' : 'Dog',
```

```
'Laptop' : 'Laptop',
'Elephant' : 'Elephant',
'Toothbrush' : 'Toothbrush',
'Baseball bat' : 'Baseball bat',
'Snowboard' : 'Snowboard',
'Train' : 'Train'}
```

**COCO to ObjectNet_D category mapping:**

```
COCO_to_ObjNet_dict = {'Baseball glove' : 'baseball_glove',
'Cup' : ['drinking_cup', 'measuring_cup'],
'bicycle' : 'bicycle',
'Tie' : 'tie',
'Banana' : 'banana',
'Tennis racket' : 'tennis_racket',
'Microwave' : 'microwave',
'clock': 'alarm_clock',
'Hair drier': 'hair_dryer',
'Dining Table' : 'coffee_table',
'mouse': 'computer_mouse',
'Remote': 'remote_control',
'Bench': 'bench',
'Backpack': 'backpack',
'Tv':'Tv',
'Keyboard' : 'keyboard',
'Cell phone' : 'cellphone',
'Chair' : 'chair',
'Bottle' : ['water_bottle', 'cooking_oil_bottle', 'beer_bottle', 'wine_bottle'],
'Umbrella' : 'Umbrella',
'Orange' : 'Orange',
'Knife' : ['butchers_knife' , 'bread_knife'],
'Vase' : 'Vase',
'Toaster' : 'Toaster',
'Bowl' : ['mixing_salad_bowl', 'soup_bowl'],
'Suitcase' : 'briefcase',
'Book' : 'book_closed',
'Laptop' : 'laptop_open',
'Baseball bat' : 'Baseball_bat'}
```

## H    DATASET STATISTICS

Some statistics of the proposed datasets are shown here. In building $COCO\_OI$ dataset, Fig. 14, we have avoided to include samples from Person, Car, and Chair categories from the OpenImages dataset. Including these categories will make the $COCO\_OI$ dataset highly imbalanced. Our code for creating this dataset ()link, however, is very general and allows discarding arbitrary object categories to make a more balanced dataset.

### H.1    DETAILS ON OBJECTNET_D DATASET

ObjectNet dataset Barbu et al. (2019), is build with the purpose of having less bias than other recognition datasets. It consists of indoor objects that are available to many people, are mobile, are not too large, too small, fragile or dangerous. This dataset is supposed to be used solely as a test set and comes with a licence that disallows the researchers to finetune models on it.

ObjectNet images are pictured by Mechanical Turk workers using a mobile app in a variety of backgrounds, rotations, and imaging viewpoints. ObjectNet contains 50,000 images across 313 categories, out of which 113 are in common with ImageNet categories. Astonishingly, Barbu et al. found that the state of the art object recognition models perform drastically lower on ObjectNet compared to their performance on ImageNet (about 40-45% drop).

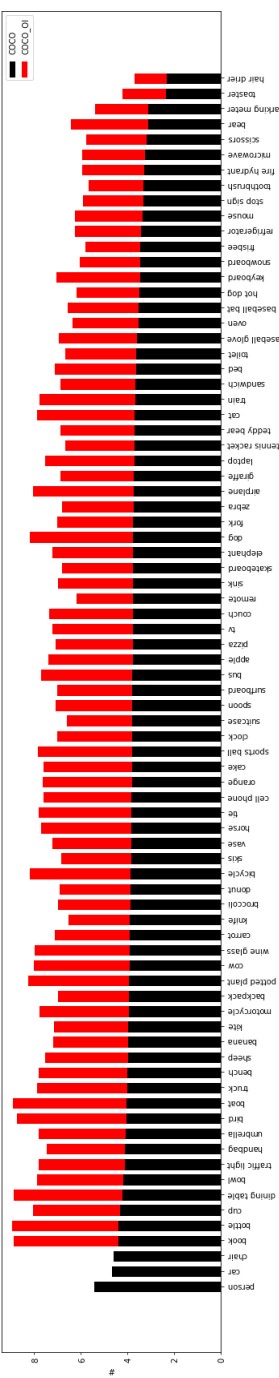

**Figure 14:** Distribution of the number of objects per category in training sets of COCO and COCO_OI datasets.

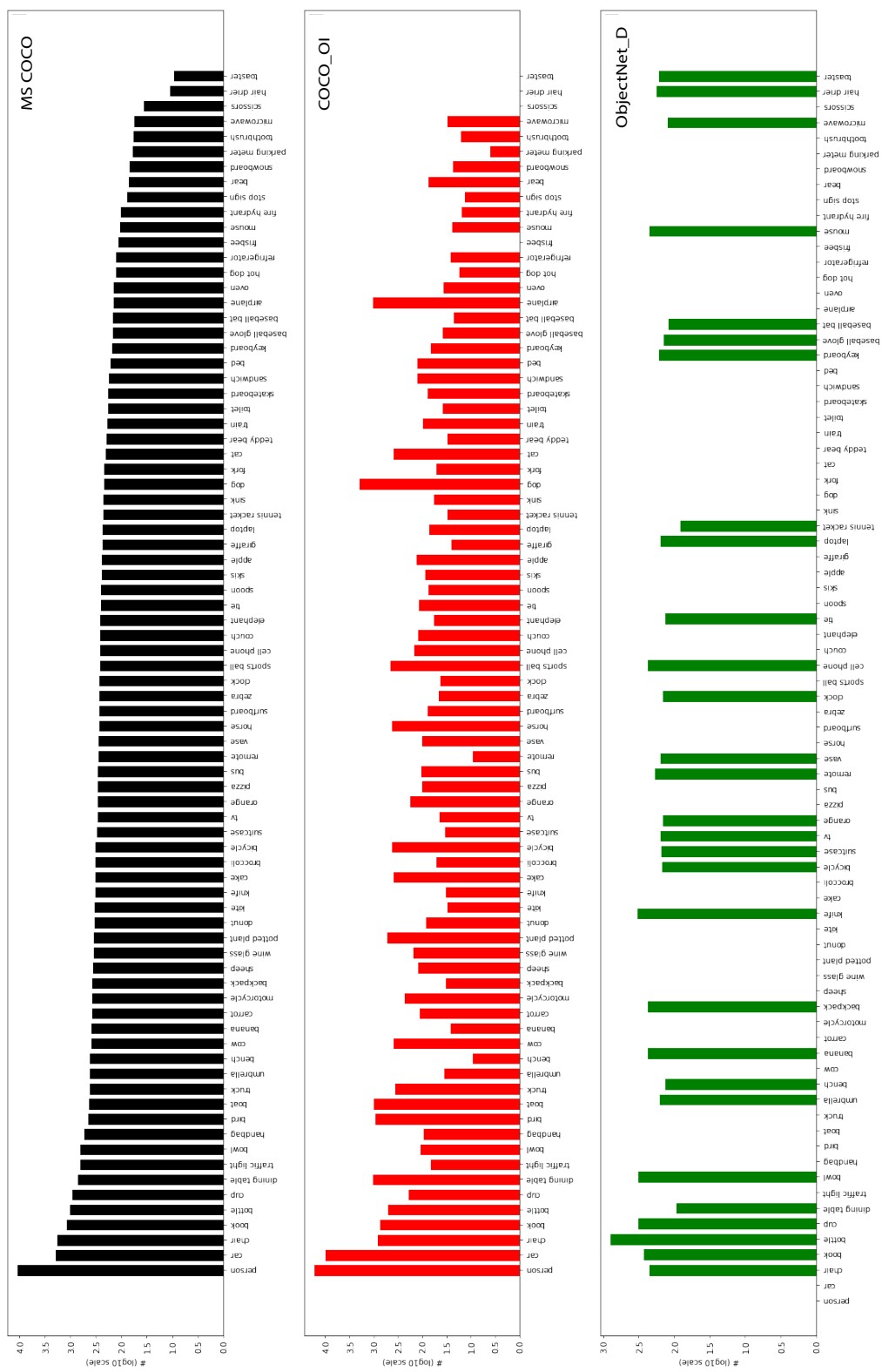

**Figure 15:** Distribution of the number of objects across the validation sets of the used datasets.

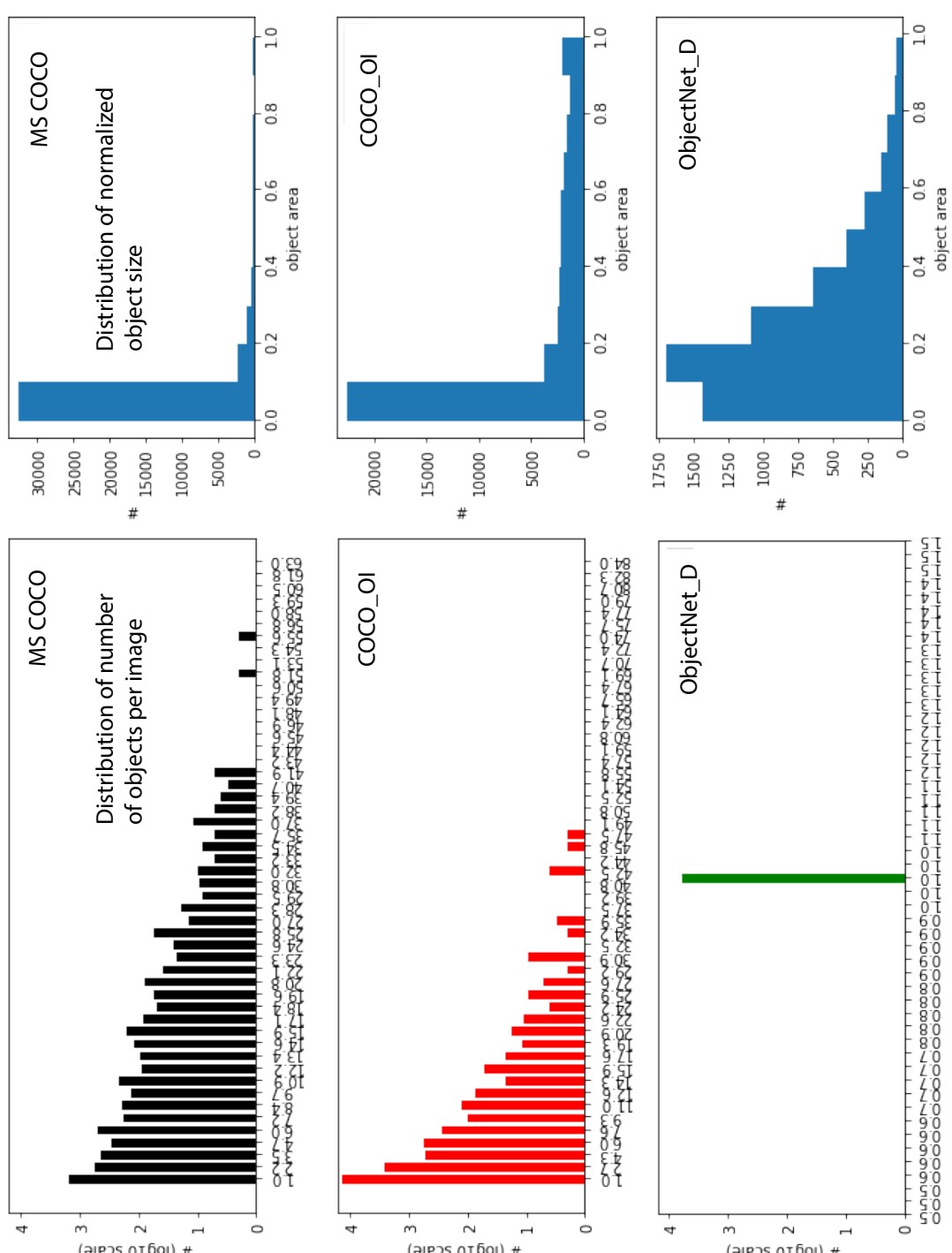

**Figure 16:** Distribution of the number of objects and normalized object size across the used datasets.

The 113 object categories in the ObjectNet dataset, overlapped with the ImageNet, contain 18,574 images in total. On this subset, the average number of images per category is 164.4 (*min*=55, *max*=284). We drew a bounding box around the object corresponding to the category label of each image. If there were multiple nearby objects from the same category (e.g., chairs around a table), we tried to include all of them in the bounding box. Some example scenes and their corresponding bounding boxes are given in Fig. 17.

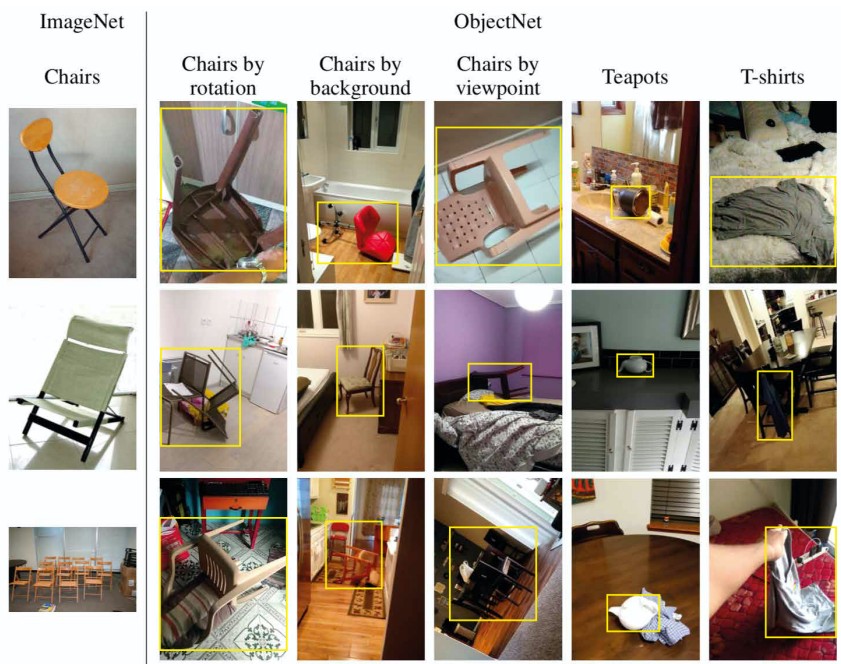

**Figure 17:** Sample images from the ObjectNet dataset along with our manually annotated object bounding boxes from `Chairs`, `Teapots` and `T-shirts` categories. The leftmost column shows three chair examples from the ImageNet dataset. ImageNet scenes often have a single isolated object in them whereas images in the ObjectNet dataset contain multiple objects. Further, ObjectNet objects cover a wider range of variation in contrast, rotation, scale, and occlusion compared to ImageNet objects (See arguments in Barbu et al. (2019)). In total, we annotated 18,574 images across 113 categories in common between the two datasets. This figure is modified from Figure 2 in Barbu et al. (2019).

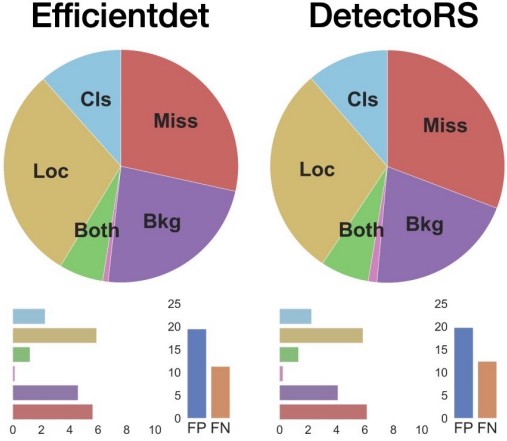

**Figure 18:** Error analysis over the ObjectNet_D dataset.

# I ANNOTATION ERRORS IN COCO DATASET

Annotation errors over COCOval2017 images, categorized in four categories, is shown in Fig. 19 (A, B, C & D). In addition, inconsistent annotations as well as some challenging images are also shown.

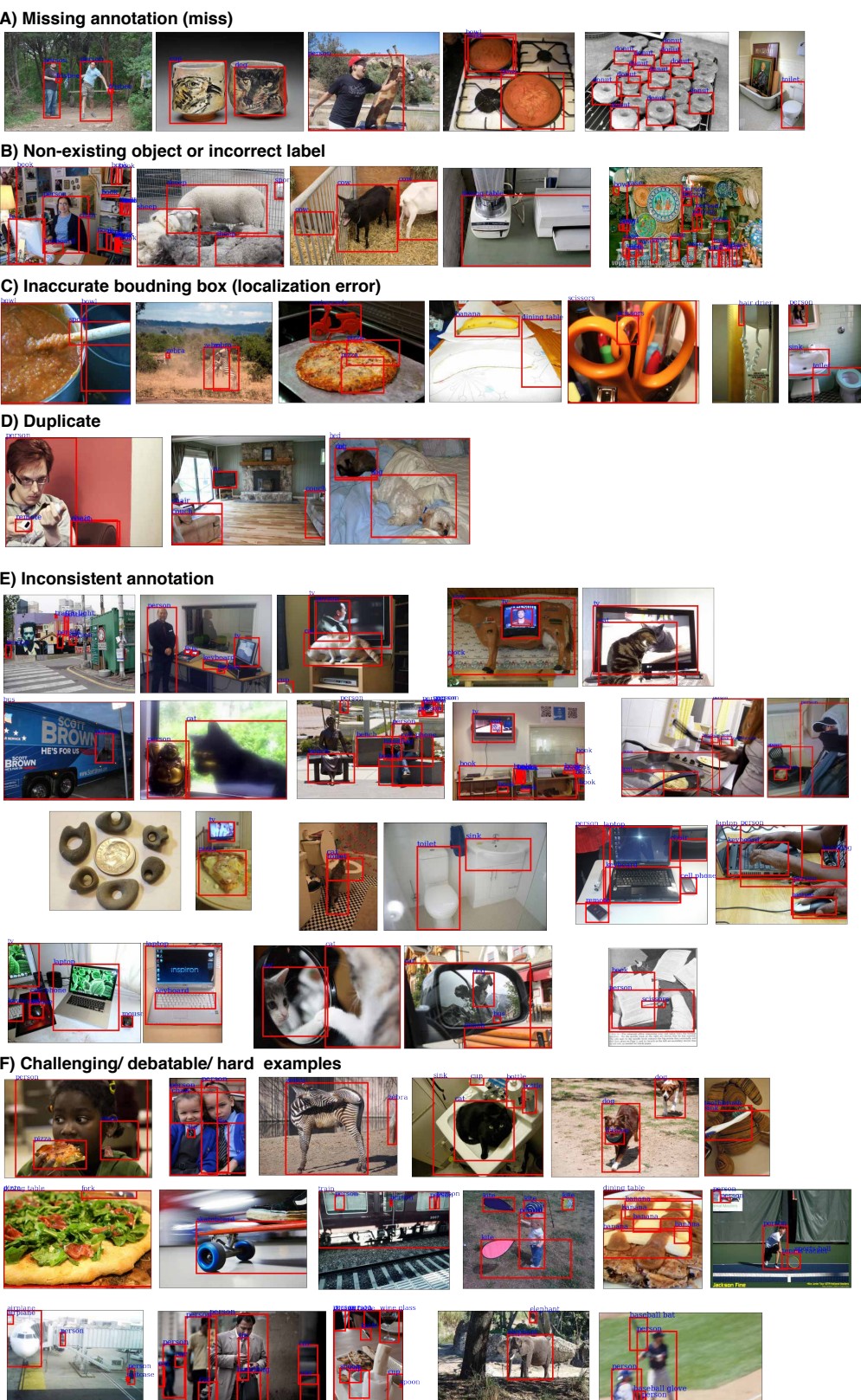

**Figure 19:** Additional annotation errors in COCOval2017.

## J    RESULTS OVER FASHION AND DOTA DATASETS

### J.1    FASHION DATASET

This dataset covers 40 categories of clothing items (39 + humans). Trainval, and test sets for this dataset contain 206,530 images (776,172 boxes) and 51,650 images (193,689 boxes), respectively. Fig. 20.A displays samples from this dataset (and also additional statistics). This is a challenging dataset since clothing items are non-rigid as opposed to COCO or VOC objects.

Results are shown in Fig. 22. The best classification accuracy on this dataset is 88.8%. The UAP is 71.2 and the AP of the best model is 59.7 (FCOS). Interestingly, FCOS performs quite close to the upper bound at IOU=0.5. Models perform better here than over VOC. The FASHION UAP is lower than VOC UAP perhaps because classification is more challenging on the former dataset. The gap between UAP and model AP here, however, is much smaller than VOC. This could be partly due to the fact that FASHION scenes have less clutter and larger objects than the VOC scenes. While per-class UAP is above the AP of the best model over all VOC classes, UAPs of five FASHION categories fall below the best model AP (*messenger bags, tunics, long sleeve shirts, blouses, and rompers*). Looking at the classification scores, we find that they have a low accuracy.

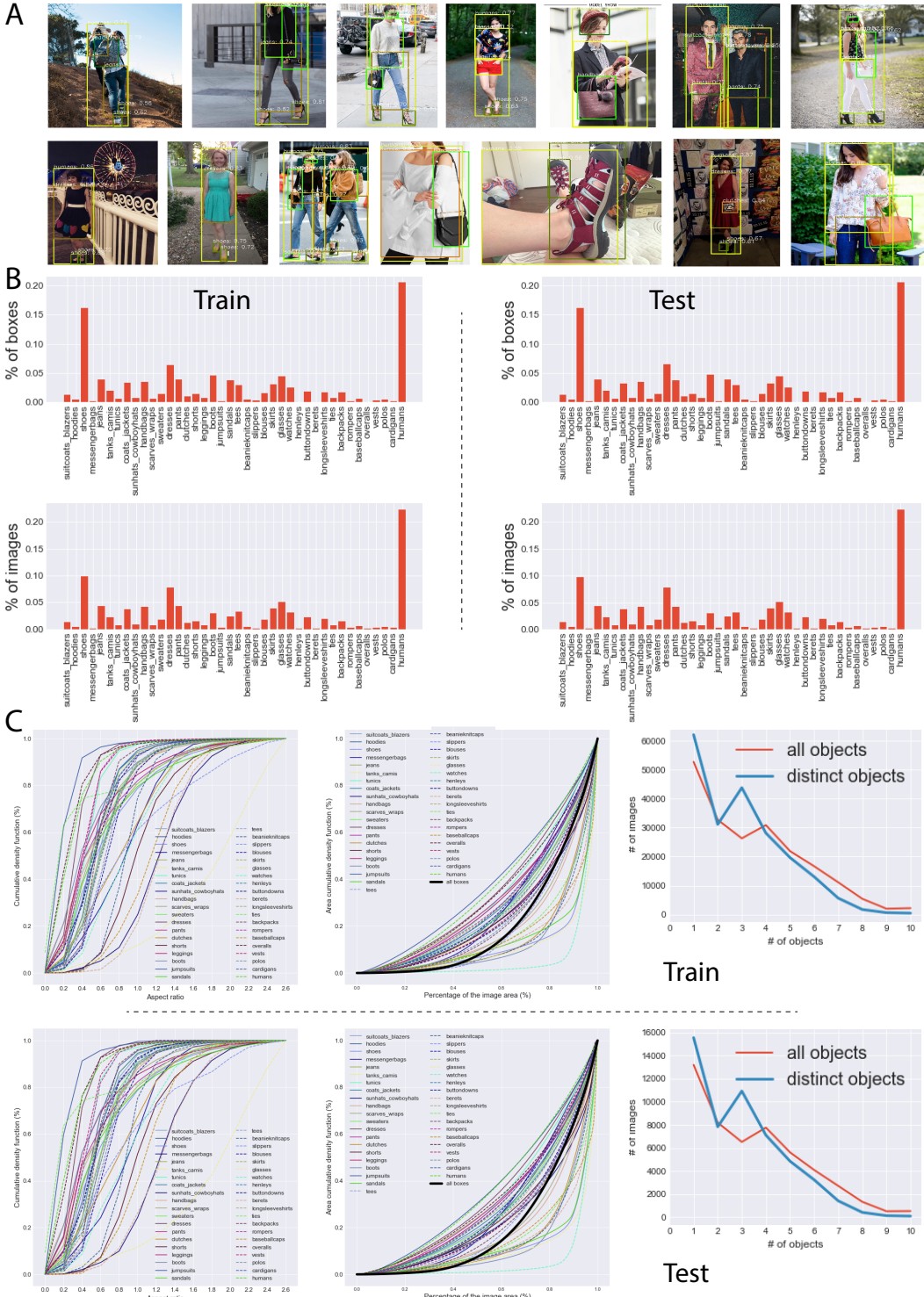

**Figure 20:** Statistics of the FASHION dataset. A) Sample images along with FCOS predictions, B) Percentage of annotated bounding boxes and images in train and test sets, and C) Aspect ratio, object size, and number of objects per image.

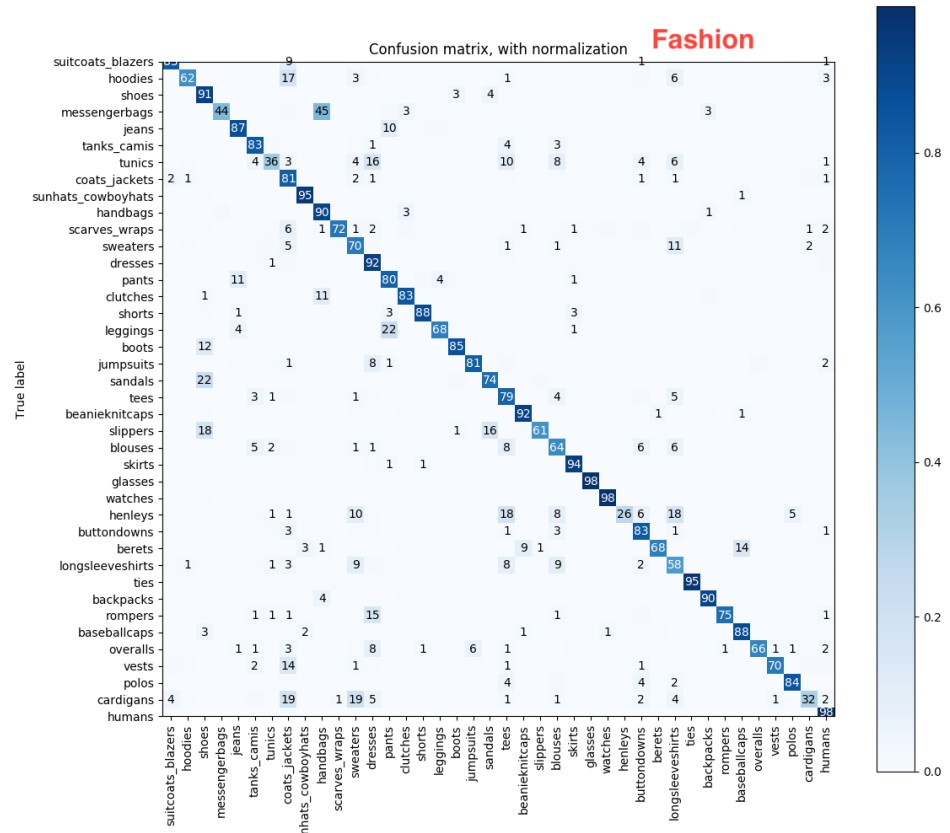

**Figure 21:** Confusion matrix of the trained classifier on the original object size over the FASHION dataset (numbers are in %).

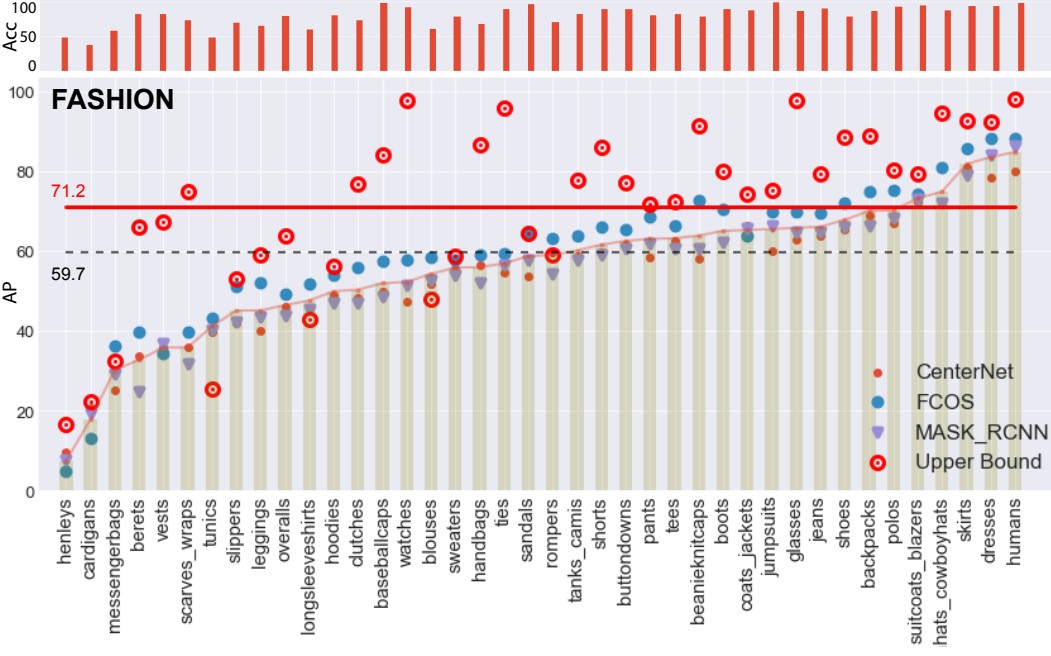

**Figure 22:** Upper bound and model APs over the FASHION dataset.

## J.2 DOTA DATASET

We are using the v1.0 set of this dataset (`https://captain-whu.github.io/DOTA/index.html`) which includes the following 15 categories: plane, ship, storage tank, baseball diamond, tennis court, basketball court, ground track field, harbor, bridge, large vehicle, small vehicle, helicopter, roundabout, soccer ball field and swimming pool. This dataset contains 2,806 images and 188,282 instances. The proportions of the training set, validation set, and testing set in DOTA-v1.0 are 1/2, 1/6, and 1/3, respectively. We are computing the UAP over the validation set of this dataset.

We are computing the UAP for the horizontal bounding box detection task on this dataset. Results are shown in Fig. 23. The trained classifier (as mentioned in the main text) scores about 89% on average over the categories. The UAP is about 88% which is higher than the AP of the best model on this dataset (about 83%). Please see `https://captain-whu.github.io/DOTA/results.html`.

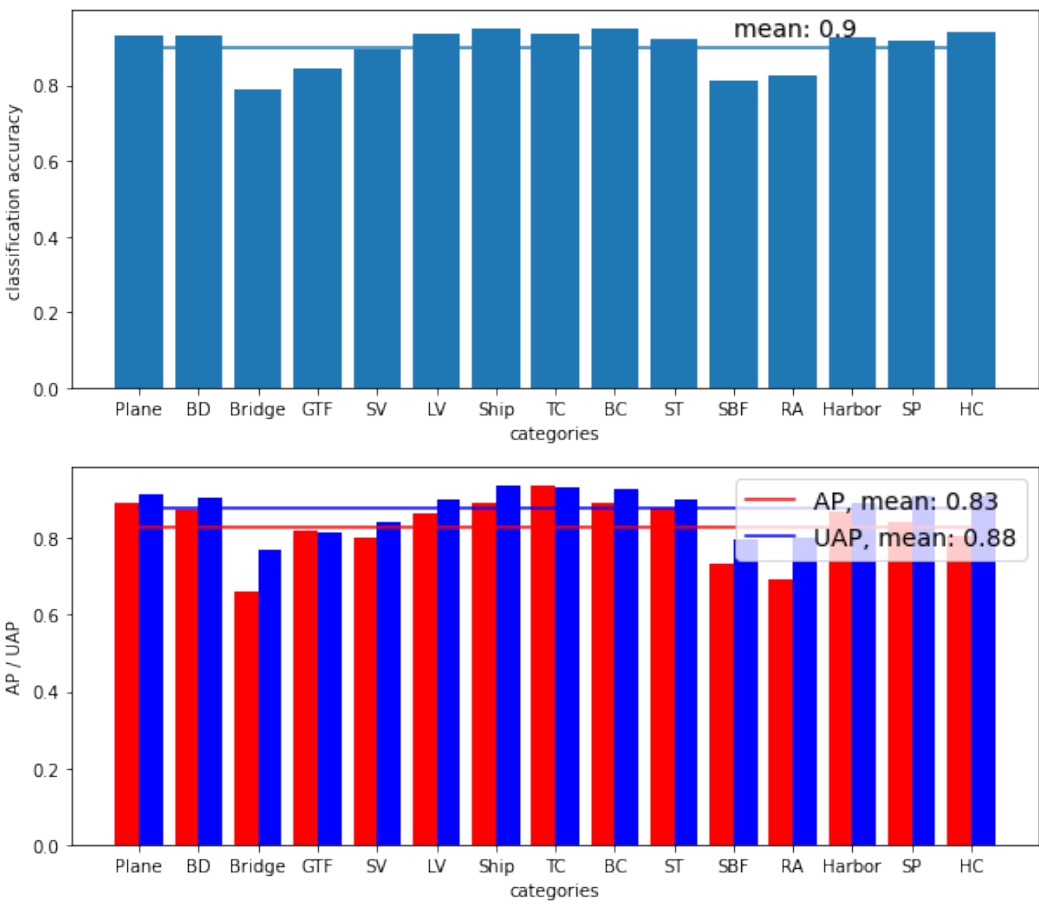

**Figure 23:** Upper bound and model APs over the DOTA dataset.

## K    COMPUTE REQUIREMENTS

All experiments were performed on a machine with 4 NVIDIA V100 GPUs and 16GB RAM.

## L    DATASET LICENSES

Below we list each dataset's license, as provided either in the paper proposing the dataset or on the dataset website. For datasets where we were unable to find a license, we list "No License."

1. MSCOCO: Creative Commons Attribution 4.0 License, `https://cocodataset.org/#termsofuse`

2. PASCAL VOC: No license specified

3. OpenImages: CC BY 4.0 license, the images are listed as having a CC BY 2.0 license `https://storage.googleapis.com/openimages/web/factsfigures.html`

4. ObjectNet: Creative Commons Attribution 4.0 with only two additional clauses: a) ObjectNet may never be used to tune the parameters of any model, and b) Any individual images from ObjectNet may only be posted to the web including their 1 pixel red border. See `https://objectnet.dev/download.html`.

