# OpenReview forum: "Breaking Beyond COCO Object Detection"
_ICLR.cc/2023/Conference — Submitted to ICLR 2023_

### Official Review · Reviewer_a5Z9 · 2022-10-25

**Confidence:** 3
**Correctness:** 3
**Technical Novelty And Significance:** 3
**Empirical Novelty And Significance:** 2
**Recommendation:** 3

**Clarity, Quality, Novelty And Reproducibility:**

* Clarity. Overall, the writing is okay but not clear enough. The organization of the paper requires a lot of back and forth redirecting to understand the concepts & settings.

* Quality. The quality of experiments looks pretty good.

* Novelty. I did not learn a lot of new insights out of this paper.

* Reproducibility. I believe the results are reproducible.

**Strength And Weaknesses:**

Strength:

+ Some of the studies are providing interesting aspects in understanding object detectors. The way of finding UAP is quite solid and experiments are comprehensively conducted on multiple datasets.

+ The authors have also spent effort in collecting new validation datasets to study the out of domain generalization performance of object detectors.

Weakness:

- The writing and presentation of this paper needs improvement. After reading the introduction, I could barely see any intuitive statement that briefly describes the results from the studies. Instead, it is full of statements that claims the results are important and interesting but without convincing me why and how. Meanwhile, I found it very hard to read Figures as they are placed in position that is far from the text. For instance, the actual text that explains Figure 1 is at Section 3.4 (page 5) but Figure 1 is at page 1.

- While UAP is claimed to study the best model's upper bound performance (assuming near-perfect localization), utilizing an outdated classifiers such as ResNet152 does not seem ideal. Wouldn't it be more interesting to fine-tune a pre-trained classifiers such as CLIP model make UAP more accurately approximating the upper bound?

- What other insights did UAP tell other than that there is a gap between the classifiers for detectors and object recognition models? I did not learn anything that is very unexpected from those studies. In fact, can such trained "upper-bound" classifiers be plugged into existing detectors to significantly improve their performance? If the answer is no, does this mean that the key challenge is to improve the localization performance?

- What is the connection between the proposed new datasets and the UAP studies in section 3? In general I see a gap between section 3 and section 4, why do we need those two complementary datasets?

- What is the role of these new validation sets comparing to the LVIS validation sets? In principle you can also use the LVIS dataset to study the out-of-domain generalization?


**Summary Of The Paper:**

Summary:

This paper studies the limitations of the COCO benchmark, raises the questions about the future directions for improving object detection systems. Particularly, this paper studies the approximated empirical upper bound performance of average precision ()UAP), and conclude that there is still a big gap between it and the best model's AP. Furthermore, this paper also introduce new validation sets to study out-of-distribution generalization performance of current object detectors. Studies were conducted to analyze the bottleneck of existing object detectors and characterize the type of errors on the new validation sets.


**Summary Of The Review:**

As pointed out in the weakness, I think this paper is a good attempt to revisit the topic of analyzing object detectors in the modern context. However, there are many issues in the execution and writing that made me to have difficulty in understanding the true insights behind. I thus could not support this paper for acceptance in its current presentation.

---

### Official Review · Reviewer_A2JS · 2022-10-25

**Confidence:** 4
**Correctness:** 3
**Technical Novelty And Significance:** 2
**Empirical Novelty And Significance:** 3
**Recommendation:** 6

**Clarity, Quality, Novelty And Reproducibility:**

The paper's idea is novel, including extensive experimental results, also code and data will be released once accepted.

**Strength And Weaknesses:**

Strengths
1. The addressing topic is of interest to a wide-scope audience.
2. The additional studies on the influence of context on detections and the experiments are interesting and comprehensive.

Weaknesses
1. The paper is hard to follow at times since all the text information and figures are not well organized and structured, and there are 35+ pages from the main paper plus the appendix, so readers tend to miss the emphasized points.
2. The paper points out that a better classification area is what researchers should focus on, but it seems to be a bit handwaving for me. It would be great if the authors could prove the conclusion, presenting us with a better object detection solution based on the findings.
3. The gap between the current detector's performance to the empirical upper bound might not hold true since the paper lacks a discussion on the most recent transformer-based solutions on the shelf now.
4. Not sure if the proposed UAP is too loose since we over-simplify the problem by decoupling the detection into localization and classification and assuming the error types from the localization are fixed.


**Summary Of The Paper:**

This work presents a method to empirically estimate the upper-bound AP (UAP) for object detection performance using a robust classifier to classify cropped bounding boxes. Specifically, the technique disentangled the detection problem into localization and classification. Assuming the object localization is correct and classification is the bottleneck, they treated the resulting AP by a trained robust classification model (ResNet152) as the empirical upper bound on detector performance. Furthermore, the paper also analyzes the few variations of their settings to check the UAP change, for instance, by adding context (enlarging/shrinking the boxes) or by adding boxes to the evaluation set that don't perfectly overlap with the ground truth. The authors conclude that the deficit in AP lies in the classification error and want to provide these insights to the community for refocusing object detection research direction in the future.




**Summary Of The Review:**

The topic of knowing the detector's gap to the upper bound performance is of interest to a wide scope of readers, however, the paper is at times hard to follow, and I do not see a concrete solution proposed based on the derived findings. Therefore, it is too soon to tell if the claimed findings are helpful to fellow researchers.

---

### Official Review · Reviewer_4FQY · 2022-11-02

**Confidence:** 3
**Correctness:** 2
**Technical Novelty And Significance:** 2
**Empirical Novelty And Significance:** 2
**Recommendation:** 3

**Clarity, Quality, Novelty And Reproducibility:**

* Clarity: The paper is not coherent in high-level logic but is clear in explaining the detailed method.
* Quality: see strengths and weakness, the paper needs some more experiments & analysis to justify the contribution
* Originality: the originality is good



**Strength And Weaknesses:**

**Strength:** the paper provides a comprehensive analysis of the experiments

**Weakness:**
* The paper is not well organized, in a way that the motivation of the proposed method & experiments is not coherent throughout the paper. (e.g. what's the connection of upper bound AP to the new proposed dataset? what's the relationship between sections 4.1 4.2 and 4.3, is the limitation of the COCO dataset (4.3) inspire the composition of the new datasets (4.1)?) The paper is disconnected in logic.
* The definition of the empirical upper bound is conditioned on knowing the exact location of the bounding box. So the gap between UAP and state-of-the-art AP largely indicates the difficulty in localization. But knowing this gap does not give us a hint about how to tackle the challenge that the current detector is facing, which makes this less exciting.  And it's debatable whether localization and recognition can be separated, this assumption needs more discussion in the paper
* What's the motivation for the new proposed data? Simply saying that the OpenImages dataset is not largely adopted is not a very convincing reason for proposing an extension for the COCO dataset. how is it solving the limitation of the COCO dataset(4.3)? It's not clear to me since there is only limited analysis on the new dataset, and there is no claim of benefit besides its OOD nature.
* The paper uses Efficientdet and DetectoRS to evaluate the difference in dataset, it would be greater to incorporate more detectors (e.g. deformable detr[1]) since different methods can have different properties when analyzing errors.

*[1] Zhu, Xizhou, et al. "Deformable DETR: Deformable Transformers for End-to-End Object Detection." International Conference on Learning Representations. 2020.*

**Summary Of The Paper:**

This paper evaluated the imperical upper bound of average precision of the COCO object detection dataset which is significantly higher than the current state of the art methods.  Two new datasets are introduced as complementary of COCO dataset. Source of error analysis showed that have state-of-the-art detectors have different behaviours on the proposed dataset.

**Summary Of The Review:**

The paper first attempts to analyze the empirical challenge in the current detection dataset, then propose two new datasets and point out the limitation of COCO. Despite the richness of the analysis, the paper is disconnected in overall logic and is not very clear in stating motivation. Overall the contribution of this paper is not very clear to me and I think a major revision/explanation is needed to justify the contribution.

---

### Official Review · Reviewer_GAsE · 2022-11-02

**Confidence:** 3
**Correctness:** 1
**Technical Novelty And Significance:** 3
**Empirical Novelty And Significance:** 2
**Recommendation:** 6

**Clarity, Quality, Novelty And Reproducibility:**

Paper is clearly written and organized, the proposed method is easy to reproduced. Technical novelty is limited.

**Strength And Weaknesses:**

+ paper is well written and easy to follow
+ the new dataset is a good addon to limited existing detection datasets.

- I think the technical contribution and novelty of this paper is limited. There are couple of papers analyzing the upper bound performance of object detectors using similar techniques.

**Summary Of The Paper:**

This paper compreehensively analyzed the upper bound performance of object detectors by partially uncovering the ground-truth labels to the models. The experiments demonstrate that the performance of current object detectors are far from perfect.

The paper also introduced a new dataset based on existing datasets (OpenImages, DOTA). The authors annotated the images from these existed datasets to create a new dataset for object detection.

**Summary Of The Review:**

Given the limited technical novelty but a new dataset, I hold a neutral attitude toward this paper.

---

### Official Review · Reviewer_5rhm · 2022-11-03

**Confidence:** 4
**Correctness:** 3
**Technical Novelty And Significance:** 3
**Empirical Novelty And Significance:** 3
**Recommendation:** 5

**Clarity, Quality, Novelty And Reproducibility:**

The paper presents a novel idea and approach. However, clarity is an issue with the current version. The authors need to address the weaknesses mentioned above. The quality of the paper can be improved by making it self-contained and not putting everything in the appendix.

**Strength And Weaknesses:**

The biggest strengths of the paper are the following:

  1. The idea of coming up with an empirical upper bound for object detection on COCO. This provides a sort of target for object detectors and also shows the limitations of existing classifiers.

  2. The analysis of the proposed UAP along with different ways of obtaining a UAP.

  3. The motivation presented in the first paragraph of section 3 is very good and the explanation is commendable.


However, the paper suffers from significant weaknesses which need to be addressed in a re-submission. In particular, I would mention the following:

  1. The paper writing needs to improve. Currently, most of the paper is in the appendices. The authors need to select the most important things and bring them in the main paper. Appendices should be only for additional information.  The paper should be self-contained and there should not be so many things in the appendices. This is unfair to other papers which had to include everything within the paper. I would recommend that the authors write the paper properly and maybe submit it as a journal paper instead of a conference with page limits.

  2. The authors use an object recognition model to classify objects. It's not clear how is this selected. The authors should provide motivation for why did they select one classifier over another. How do you determine which object detector's classification head works best? Doesn't the UAP depend on the selected classifier? And if yes, how much?

  3. The proposed UAP assumes that the ground-truth bounding boxes are given and a classifier needs to be trained. Does this assume that the bigger problem is localization? Can there be another empirical UAT where the classes can be taken from GT but the model needs to generate boxes.

  4. Though the authors mention that they include the most recent models in their analysis, I would encourage them to include transformer-based models like DETR/MDETR as well.

  5. Section 3.2 needs more detail. It's not clear to me how exactly is the classifier trained and how the GT boxes are used.

  6. In Table 1, please mention what is "Acc." in the caption.

**Summary Of The Paper:**

The paper attempts to present an empirical upper bound for COCO and show that the existing approaches to object detection are far from this upper bound. The authors further propose two datasets complementary to COCO. The authors show that object detection models behave differently on the proposed datasets compared to COCO.

**Summary Of The Review:**

The idea of analyzing the performance of object detection models using an empirical upper bound is novel and interesting. However, the paper needs to be written better - in particular, making it self-contained by selecting the most important points, and clarifying some of the details of training and using the classifier. Therefore, I am recommending rejecting the paper. I would recommend that the paper include the most important details and experiments from the appendix into the main paper and submit to another venue - maybe a journal.

---

### Decision · Program_Chairs · 2023-01-20

**Decision:**

Reject

**Justification For Why Not Higher Score:**

While the paper provides some interesting analysis, its lacking and confusing exposition as well as lack of clear (novel) insights that would make actionable changes to object detection makes it no ready for publication.

**Justification For Why Not Lower Score:**

N/A

**Metareview: Summary, Strengths And Weaknesses:**

Paper characterizes the upper bound performance and studies the limitations of the COCO benchmark. In doing so, it raises questions regarding future directions for improving object detection. Paper received five reviews from expert reviewers, with slightly divergent scores:

- 1 x marginally below the acceptance threshold
- 2 x marginally above the acceptance threshold
- 2 x reject, not good enough

Notably, none of the reviewers strongly argued for acceptance, while two reviewers did strongly argue for the rejection of the paper. Further review of [GAsE] was lacking detail and was largely discounted by the AC.

On the positive side, all reviewers appreciated the analysis and overarching goal of characterizing the upper bound and using it to improve future performance. That said, there was also a number of significant concerns with the work. Mainly, (1) lacking writing and exposition that made it hard to understand the key findings [5rhm, 4FQY, A2JS, a5Z9], (2) lack of comparisons to the most recent transformer-based architectures [5rhm,4FQY], (3) lack of "surprising" findings that transcend our current understanding [4FQY, a5Z9], e.g., the gap between UAP and state-of-the-art AP largely indicates the difficulty in localization, which is well known in the community.

Authors provided a through rebuttal, but it largely addressed (2) by adding experiments with DETR baselines. Other concerns do remain and are significant enough to warrant Rejection of the paper at this time. AC has reviewed the original reviews, rebuttal and the paper itself and concurs with majority reviewer opinion that paper is not ready for publication. While it does present some interesting analysis, it needs further work to appropriately position itself and to organize and potentially draw more insightful conclusions.